# Nutrient supply controls the linkage between species abundance and ecological interactions in marine bacterial communities

Tianjiao Dai[1,2], Donghui Wen[2 ✉], Colin T. Bates[3], Linwei Wu [3], Xue Guo [1], Suo Liu[1], Yifan Su[1], Jiesi Lei[1], Jizhong Zhou [3,4,5] & Yunfeng Yang [1 ✉]

Nutrient scarcity is pervasive for natural microbial communities, affecting species reproduction and co-existence. However, it remains unclear whether there are general rules of how microbial species abundances are shaped by biotic and abiotic factors. Here we show that the ribosomal RNA gene operon (*rrn*) copy number, a genomic trait related to bacterial growth rate and nutrient demand, decreases from the abundant to the rare biosphere in the nutrient-rich coastal sediment but exhibits the opposite pattern in the nutrient-scarce pelagic zone of the global ocean. Both patterns are underlain by positive correlations between community-level *rrn* copy number and nutrients. Furthermore, inter-species co-exclusion inferred by negative network associations is observed more in coastal sediment than in ocean water samples. Nutrient manipulation experiments yield effects of nutrient availability on *rrn* copy numbers and network associations that are consistent with our field observations. Based on these results, we propose a "hunger games" hypothesis to define microbial species abundance rules using the *rrn* copy number, ecological interaction, and nutrient availability.

[1] State Key Joint Laboratory of Environment Simulation and Pollution Control, School of Environment, Tsinghua University, Beijing, China. [2] College of Environmental Sciences and Engineering, Peking University, Beijing 100871, China. [3] Institute for Environmental Genomics and Department of Microbiology and Plant Biology, University of Oklahoma, Norman, OK, USA. [4] School of Civil Engineering and Environmental Sciences, University of Oklahoma, Norman, OK, USA. [5] Earth and Environmental Sciences, Lawrence Berkeley National Laboratory, Berkeley, CA, USA. ✉email: dhwen@pku.edu.cn; yangyf@tsinghua.edu.cn

Microbial communities typically show a skewed abundance distribution, where a few species are highly abundant, and a long tail of other species are in low abundance[1–3]. This phenomenon provokes an interest in investigating the rare biosphere[1] for their importance regarding species diversity[4,5], contribution to community dynamics[6], and ecosystem functions[4,7]. Most prokaryotic species propagate by clonal replication, whose abundance profiles rely on various nutrient demands because most microorganisms are auxotrophs. As a result, some species are persistent members of the rare or abundant biosphere, and others are variable in abundance[3]. Different abundance profiles have led to a hypothesis on discriminating life-history strategies for the rare and abundant organisms[3,8], referring to multifaceted traits related to growth rates and nutrient utilization efficiency[9]. The fast-growing organisms produce many ribosomes for protein synthesis to keep up the growth rate, but at a tradeoff of compromised metabolic enzyme production and nutrient utilization efficiency[10–12]. On the contrary, slow-growing organisms allocate more energy to metabolic enzymes when nutrient is scarce[11,13]. The growth rate is constrained by carbon (C): nitrogen (N): phosphorus (P) stoichiometry, termed as the "growth rate hypothesis"[14–16], which has been observed in animals[17], plants[18], and zooplankton[16]. The fast-growing organisms rely on external nutrients to support the phosphorus (P)-rich ribosomal RNA production, resulting in high abundance when there is ample nutrient supply in the environment[14].

However, abundance profiles are subject to various interspecies interactions that affect individual organism growth and loss rates[2,3]. Bacterial communities constitute "social networks" in which the members interact with each other in various ways, including competition for nutrients, cooperation by cross-feeding, communication via secretion, and detection of extracellular substances[19]. In addition, organisms may also indirectly affect other community members by modifying their environment, termed "niche construction theory"[20]. For example, excretion of secondary metabolites by actively growing species could change environmental conditions, influencing the growth of other organisms and shifting relative abundance levels[21]. Theoretically, biotic factors (e.g., growth traits and ecological interactions) and abiotic factors (e.g., nutrient supplies) intertwine to affect individual species abundance, which determines community composition. Nonetheless, whether there are universal rules of biotic factors and nutrient availability in shaping species abundance in natural bacterial communities remains unclear.

Here, we compared bacterial community composition in coastal sediments across Asia, Australia, Europe, North America, and South America (nutrient-rich environments due to riverine deposition and anthropogenic-induced eutrophication, 243 samples) to that of the free-living fraction in the ocean water (nutrient-scarce environment, 139 samples collected in the *Tara* Oceans Project) (Supplementary Data 1). We also carried out microcosm studies to experimentally examine how nutrient availability affects natural bacterial communities. The ribosomal RNA gene operon (*rrn*) copy number in bacterial genomes is a phylogenetically conserved trait at the genus and species levels[22], which can predict the growth rate and nutrient utilization efficiency for individual organisms well[10,11,23]. Accordingly, we examined whether the average *rrn* copy number of community members was positively correlated with environmental nutrient contents, which was verified in natural[24] and engineered systems[25,26]. We also mathematically modeled a microbial community as a network to explore potential ecological interactions with nutrient availability. Based on those analyses, we aimed to test two hypotheses: (i) *rrn* copy numbers in rare and abundant biospheres differ in nutrient continuums (i.e., from sediments to ocean water and microcosm studies), contingent on nutrient availability; and (ii) potential ecological interaction is a generalizable mechanism in microbial responses to nutrient availability.

## Results

**The *rrn* copy numbers of individual OTUs in abundant, intermediate, and rare biospheres.** We classified the abundant, intermediate, and rare biospheres in coastal sediment and ocean water samples based on OTUs' abundance and occurrence frequency (see Fig. 1 for detail). Less than 1.46% of OTUs in the datasets were classified as the abundant biosphere, whereas 61.40% ~ 98.51% of the OTUs belonged to the rare biosphere, and the rest OTUs were the intermediate biosphere between rare and abundant biospheres (Supplementary Table 1).

We estimated bacterial OTU's *rrn* copy number based on the *rrn*DB database[27]. There were significant differences in OTUs' *rrn* copy numbers among the abundant, intermediate, and rare biospheres for most coastal sediment and ocean water communities (Fig. 1b and Supplementary Table 1). The OTUs' *rrn* copy numbers decreased from abundant to the rare biosphere in most coastal sediments except the coastal Mediterranean and coastal Sydney sediments, but increased in the global ocean waters (Fig. 1b and Supplementary Table 1). Compared to the coastal sediments, the OTUs' *rrn* copy numbers for the abundant biosphere in ocean water was significantly lower (Supplementary Fig. 1a), but that for the rare biosphere was significantly higher (Supplementary Fig. 1b). The abundant biosphere in the coastal sediments was frequently observed to include *Bacillales* (6~11 *rrn* copies; Hangzhou Bay and Mission Bay), *Clostridiales* (5 *rrn* copies; coastal Mediterranean), *Desulfobacterales* (4 *rrn* copies; Plymouth Harbor, coastal Sydney, and the coastal Mediterranean), *Alteromonadales* (5 *rrn* copies; the Gulf of Mexico and coastal Sydney), *Rhodobacterales* (4 *rrn* copies; coastal Mediterranean), *Flavobacteriales* (3 *rrn* copies; coastal Sydney), *Rhodospirillales* (4 *rrn* copies; the Gulf of Mexico), and *Oceanospirillales* (5 *rrn* copies; the Gulf of Mexico). In the global ocean water, 71 out of the 105 OTUs belonging to the abundant biosphere were SAR11, which has a single *rrn* copy. To assess how SAR11 and other OTUs with a single *rrn* copy affect *rrn* copy numbers, we removed those OTUs from the datasets. We showed that the *rrn* copy number of the abundant biosphere in ocean water was only significantly lower than the sediments in Mission Bay and the Gulf of Mexico (Supplementary Fig. 1c). In contrast, that of the rare biosphere was still significantly higher than the sediment samples (Supplementary Fig. 1d), indicating that single *rrn* copy OTUs in ocean water contributed the most to low *rrn* copy numbers of the abundant biosphere.

**The community-level *rrn* copy number.** We calculated the community-level *rrn* copy number as abundance weighted average *rrn* copy number throughout each community member, as described in previous studies[24,25,28]. The community-level *rrn* copy number for coastal sediments across the globe averaged 2.74 ± 0.06, ranging widely from 1.59 to 7.61. They were higher in Mission Bay (3.52 ± 0.25), Hangzhou Bay (3.18 ± 0.17), and the Gulf of Mexico (2.79 ± 0.06) than the other sediments. The community-level *rrn* copy number in global ocean water was higher in deep than surface ocean water, which could be attributed to lower SAR11 abundance in deep ocean water (Supplementary Fig. 2). However, it averaged 1.54 ± 0.01 with a range between 1.39 and 2.53 (Fig. 1c), which was still significantly ($P < 0.001$, ANOVA) lower than sediments. The abundance unweighted community-level *rrn* copy numbers in ocean water were also lower than those in sediments (Supplementary Table 1), suggesting that ignoring taxon abundance will not affect our results. By calculating the mean pairwise distance (MPD) of OTUs across the phylogenetic tree, we found that the

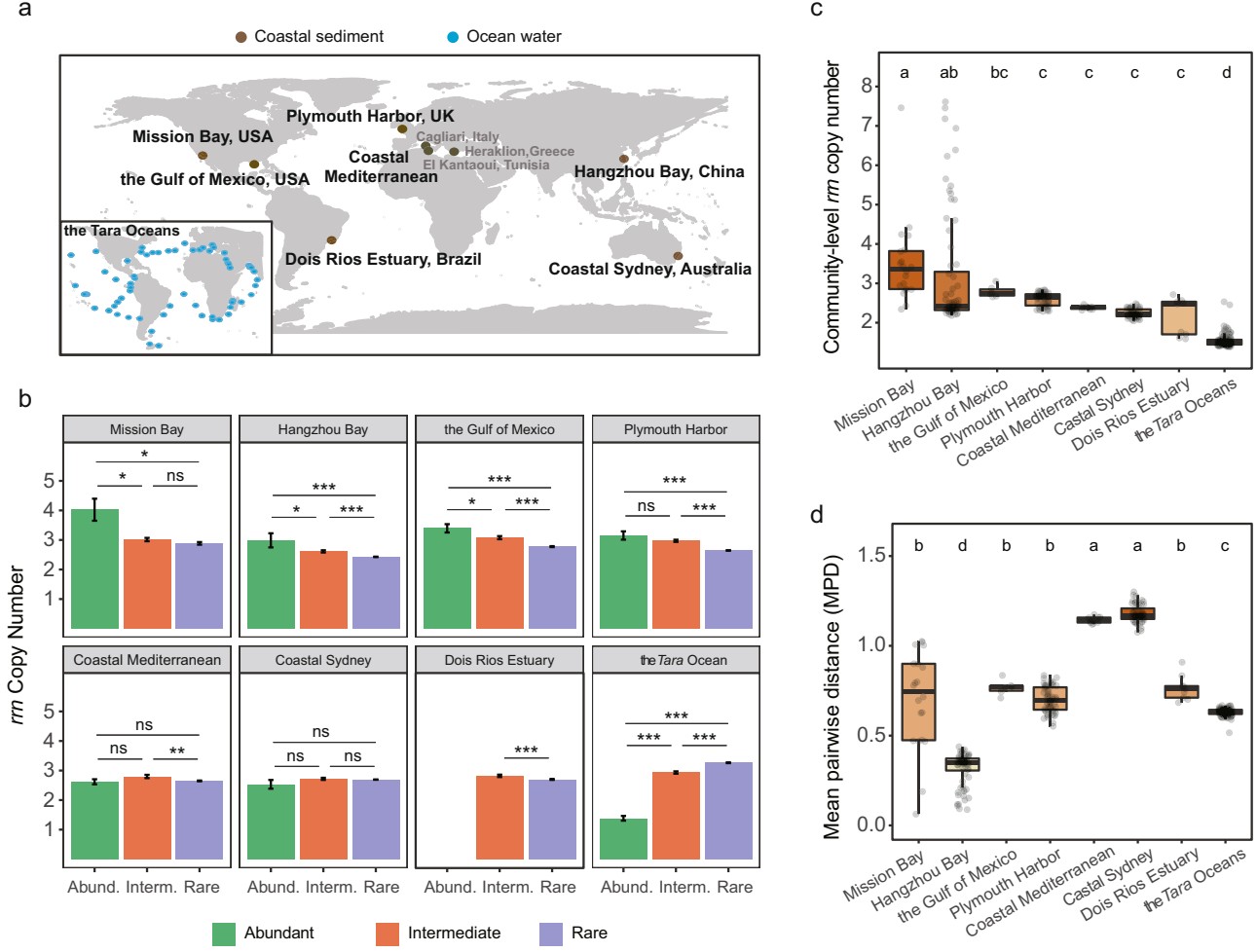

**Fig. 1 The *rrn* copy numbers of bacterial OTUs in coastal sediment and global ocean water. a** Maps of the locations where coastal sediment and ocean water bacterial communities were examined in this study. **b** OTUs' *rrn* copy number of the abundant, intermediate, and rare biospheres. The OTUs were classified based on their relative abundance and occurrence frequency: (i) abundant biosphere, OTUs with relative abundance ≥0.1%[53] in >50%[26] samples, and occurred in more than 80%[26,54] samples; (ii) rare biosphere, OTUs with relative abundance <0.1% in all samples; (iii) intermediate biosphere, OTUs other than those belonging to the rare biosphere or abundant biosphere. The bars indicate mean values and the error bars indicate standard errors. The sample sizes and detailed summary statistics are listed in Supplementary Table 1. Unadjusted *P* values of the one-way ANOVA followed by LSD test are labeled as *** when $P < 0.001$, ** when $P < 0.01$, * when $P < 0.05$, and ns when $P > 0.05$ (not significant). No OTU was classified as abundant biosphere in the dataset of Dois Rios Estuary so the bar is missed. **c**, **d** The community-level *rrn* copy numbers and phylogenetic diversity of coastal sediment and seawater and seawater bacterial communities. In the boxplots of panels (**c** and **d**) hinges indicate the 25th, 50th, and 75th percentiles, whiskers indicate 1.5 × interquartile ranges, and dots indicate values of individual samples. Lowercase letters above the bars indicate significant differences (adjusted $P < 0.05$ by Bonferroni method, one-way ANOVA followed by LSD test). The sample sizes in panels (**c** and **d**) are as follows: Mission Bay, 20; Hangzhou Bay, 72; the Gulf of Mexico, 6; Plymouth Harbor, 66; coastal Mediterranean, 11; coastal Sydney, 60; Dois Rios Estuary, 9; the *Tara* Oceans, 139. Source data for panels (**b**, **c**, **d**) are provided in the Source Data File.

communities with the highest diversity (i.e., the coastal Mediterranean and coastal Sydney in Fig. 1d) did not have the highest or lowest community-level *rrn* copy numbers (Fig. 1c), suggesting OTUs in these communities covered a broader phylogenetic lineage.

The community-level *rrn* copy number was positively correlated with all measured nutrients (r = 0.410~0.782, *P* < 0.013, Pearson's correlation) for both coastal sediment and ocean water samples (Supplementary Table 2). Partial Mantel tests confirmed significant linkages between community-level *rrn* copy number and nutrients (r = 0.130~0.602, *P* < 0.029) when differences in the underlying phylogenetic structure were controlled (Supplementary Table 3). When calculating the *rrn* copy numbers based on abundant, intermediate, and rare biospheres, we found that their correlations with nutrients varied substantially (Supplementary Table 2). In Hangzhou Bay, the *rrn* copy numbers of the abundant

and intermediate biospheres showed stronger and more significant positive correlations with N and P than that of the rare biosphere. In contrast, the *rrn* copy numbers of the rare and intermediate biospheres in global ocean water were more strongly correlated with N and P than that of the abundant biosphere (Supplementary Table 2).

**Network associations.** To evaluate potential ecological interactions among bacterial members of a community, we generated random matrix theory (RMT)-based association networks[29] for bacterial communities in the sediments of Hangzhou Bay, Plymouth Harbor, and coastal Sydney, as well as in the global ocean water (Fig. 2a) because those datasets are of sufficient sample sizes (at least 60) to allow reliable network analyses. All the networks displayed typical properties of complex systems, including scale-free, small-world, and modular characteristics.

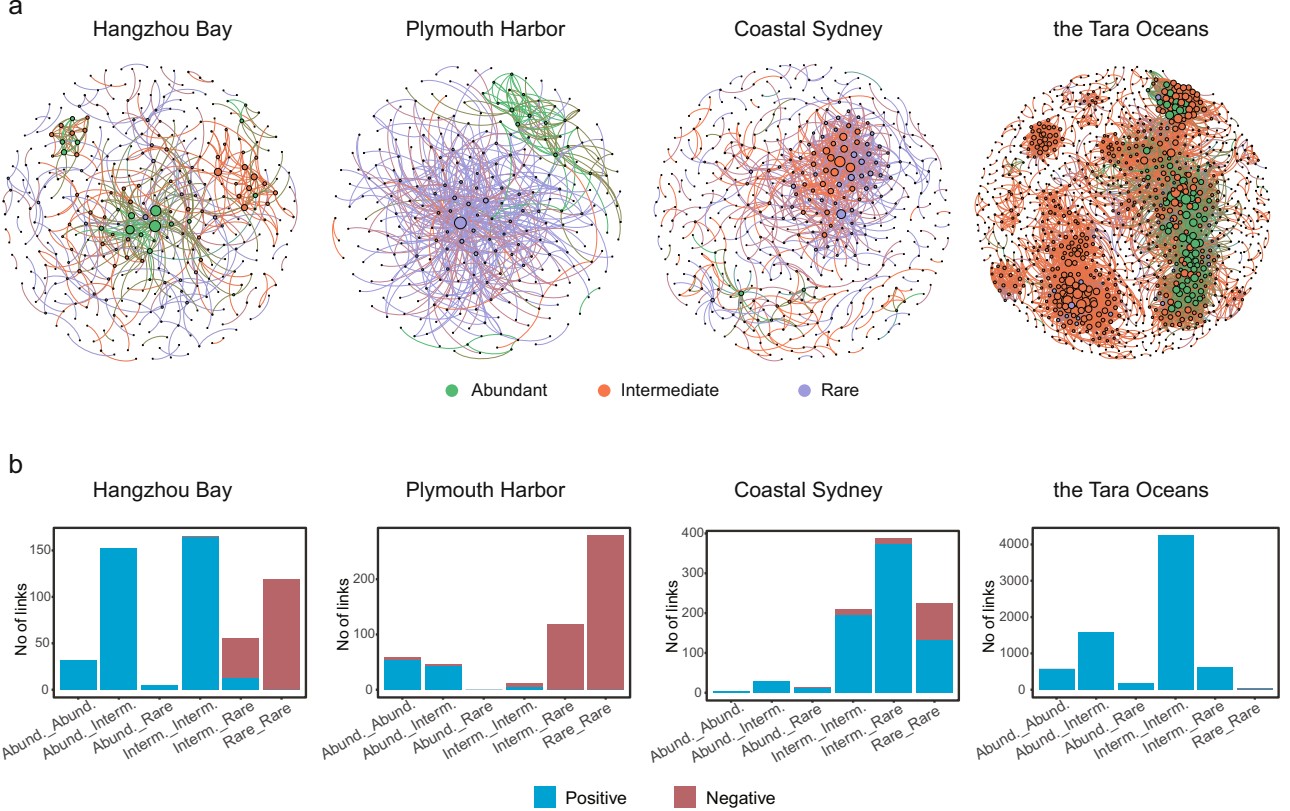

**Fig. 2 Network associations of bacterial OTUs in coastal sediment and global ocean water. a** The networks represent random matrix theory-based correlation models derived from all samples for each dataset, where nodes represent OTUs, and links between the nodes represent significant correlations. The node size is proportional to its degree, and the color indicates the abundance category (i.e., abundant, intermediate, and rare biospheres). The sample sizes for network analysis are as follows: Hangzhou Bay, 72; Plymouth Harbor, 66; coastal Sydney, 60; Dois Rios Estuary, 9; the *Tara* Oceans, 139. **b** A summary of the network links by the abundance categories of the associated OTUs. Source data are provided in the Source Data File.

They exhibited non-random features by showing significantly different topological indices from their corresponding random networks (Supplementary Table 4).

We calculated the positive and negative associations (i.e., network links) for each network to examine inter-species co-existence and co-exclusion patterns. We observed a much higher proportion of negative associations for bacterial communities in coastal sediment (Hangzhou Bay, 31.13%; Plymouth, 80.35%; coastal Sydney, 14.38%) than that in the global ocean water (0.04%) (Fig. 2b). Although the coastal samples were distinct in bacterial community composition, most of the negative associations in coastal sediments were within the rare biosphere or between the rare and intermediate biosphere (Fig. 2b). In sharp contrast, the associations within the abundant biosphere were predominantly positive.

**Validating the effects of nutrient supply on bacterial communities.** We examined the impact of nutrient availability on bacterial community assembly by manipulating $NH_4^+$-N and $PO_4^{3-}$-P concentrations in microcosms with sediments collected from Hangzhou Bay. Specifically, we added $5\,mg\,L^{-1}$ $NH_4^+$-N and $0.5\,mg\,L^{-1}$ $PO_4^{3-}$-P (denoted as low nutrient supply), and $50\,mg\,L^{-1}$ $NH_4^+$-N and $5.0\,mg\,L^{-1}$ $PO_4^{3-}$-P (denoted as high nutrient supply), whereas no nutrients were added in the control microcosms. The bacterial diversity was reduced intensively in all microcosms at the beginning three days but recovered gradually over time, both taxonomically and phylogenetically (Supplementary Fig. 3). After 28 days, bacterial diversities under low and high nutrient supply were comparable ($P = 0.12$), but significantly ($P < 0.05$) higher than that in the control

microcosms (Supplementary Fig. 3). The Principle coordinates ordination analysis (PCoA) based on Bray-Curtis distances revealed divergent succession trajectories for bacterial communities under nutrient supply conditions (Supplementary Fig. 4), with significant ($P < 0.001$, Adonis) effects of both time and nutrient supply (Supplementary Table 5).

We assigned the OTUs into the rare, intermediate, and abundant biospheres for each of the bacterial communities under the three nutrient supply conditions using the same criteria as in Fig. 1 (Supplementary Table 6). The *rrn* copy number for the abundant and intermediate biospheres in the control microcosms were comparable ($P = 0.61$) but higher than those in the rare biosphere ($P < 0.001$, Fig. 3a). Under low and high nutrient supply conditions, the *rrn* copy number decreased from abundant, intermediate to the rare biosphere ($P < 0.05$ in all comparisons, Fig. 3a). The strength of the correlations between the community-level *rrn* copy number and nutrients increased remarkably with high nutrient supply (Supplementary Table 7), particularly in the abundant biosphere and, to a lesser extent, the rare biosphere (Fig. 3b). We constructed individual association networks for control, low, and high nutrient supply samples. Compared to the control group, the OTUs were more densely linked when nutrients were supplied (Fig. 4), with higher average degrees (Supplementary Table 8) and proportions of negative associations (Supplementary Table 9). More than half of the associations (60.7 ~ 73.6%) among the rare biosphere OTUs were negative in the three networks. With a few exceptions, the proportions of negative associations increased with nutrient supply for most of the associated OTUs pairs, i.e., abundant-abundant, abundant - intermediate, abundant - rare,

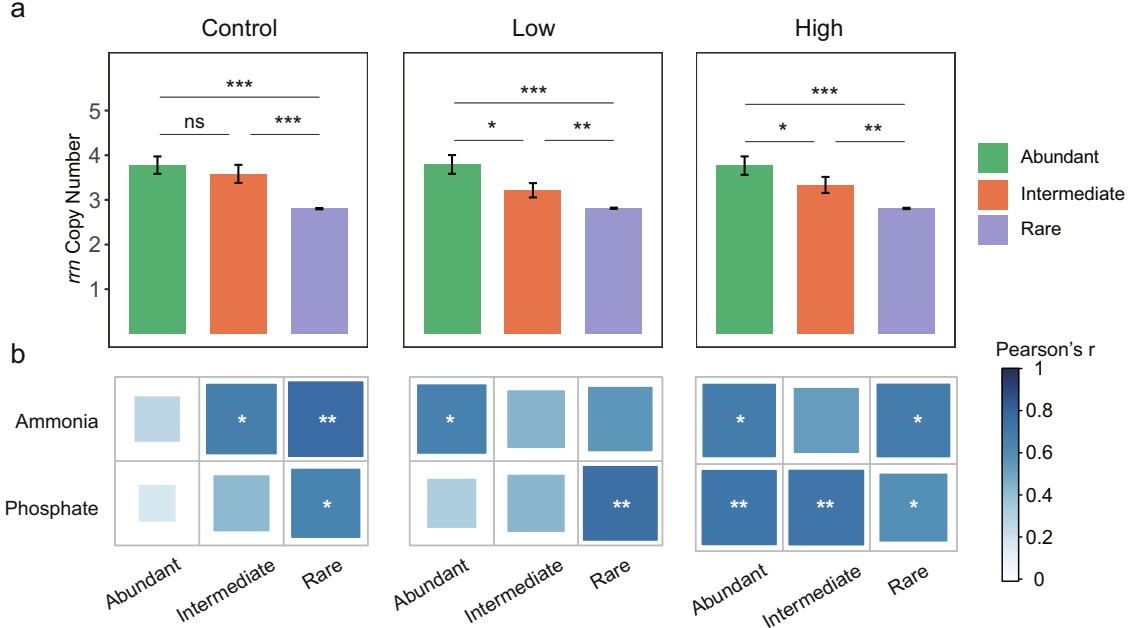

**Fig. 3 The effects of nutrient supply on *rrn* copy numbers. a** Average *rrn* copy numbers of bacterial OTUs in the abundant, intermediate, and rare biosphere biospheres in microcosms. The bars indicate mean values and the error bars indicate standard errors. Unadjusted *P* values of one-way ANOVA followed by LSD test are labeled as *** when $P < 0.001$, ** when $P < 0.01$, * when $P < 0.05$, and ns when $P > 0.05$ (not significant). The sample sizes (i.e., number of OTUs) are listed in Supplementary Table 6. The OTUs were classified based on their abundance variations over 28 days, using the same criteria as Fig. 1. **b** The heatmaps indicate two-sided Pearson's correlation coefficients between nutrient concentrations and community-level *rrn* copy number for bacterial abundant, intermediate, and rare biospheres. The *P* values of Pearson's correlation coefficients ($n = 15$ biologically independent samples for each correlation test) are labeled as *** when $P < 0.001$, ** when $P < 0.01$, and * when $P < 0.05$. Source data are provided in the Source Data File.

and intermediate - rare (Supplementary Table 9), suggesting that nutrient supply induced negative associations among community members.

## Discussion

The analyses of global marine bacterial communities allow for exploring generalizable ecological rules beyond what could be observed in individual studies. Here, we show that the bacterial genomic trait of *rrn* copy number, environmental nutrient supply, and potential ecological interactions are among the core drivers of marine bacterial community assembly. By considering the general bacterial genomic trait of *rrn* copy number and incorporating the local environmental nutrient supply along with potential ecological interactions, we propose a hypothesis to explain marine bacterial community assembly and taxon abundance. We assert this hypothesis by combining several existing theories/hypotheses such as environmental selection theory[30], growth-rate hypothesis[18], evolutionary game theory[31], and niche construction theory[20], to propose what we dub as the "hunger games" hypothesis to highlight interplays between potential ecological interactions and nutrient supplies in shaping bacterial communities. Our hypothesis contains two rules: (1) the species abundance profile of natural bacterial communities is affected by the genomic trait *rrn* copy number, wherein high *rrn* copy number is favored in copiotrophic environment, but low *rrn* copy number is favored in the oligotrophic environment; and (2) cooperation prevails in natural bacterial communities, but competition is more frequent in the copiotrophic environment. We generate a conceptual diagram to summarize it (Fig. 5). We also elaborate on the "hunger games" hypothesis below.

(1) Nutrient-rich environments favor fast-growing bacteria, whereas nutrient scarcity selects bacteria with efficient nutrient utilization. In pure cultures, it was shown that *rrn* copy number is a reliable proxy for bacterial adaptation to nutrient availability[11,12].

Here, we show that it is also true in free-living marine bacterial communities (Figs. 1, 3 and Supplementary Table 2). However, the particle-associated microbial communities have a quite different pattern where copiotrophs such as *Alteromonadales*, *Oceanospirillales*, *Flavobacteriales*, and *Rhodospirillales*, are more abundant, which have higher growth rates and *rrn* copy numbers than free-living microbes[9] and are also commonly found in coastal sediments as shown in this study. This is similar to the decrease of oligotrophs (e.g., SAR11) and enrichment of copiotrophs in deep ocean water (Supplementary Fig. 2), wherein the organic matter availability is higher due to particle deposition[32]. Furthermore, the *rrn* copy number is different among the rare, intermediate, and abundant biospheres, relying on nutrient conditions (Figs. 1b, 3a). This finding advances the life-history strategy hypothesis for rare and abundant biospheres[3,8] by pinpointing the *rrn* copy number as a genomic feature related to species reproduction success. Since ribosomes are P-rich, organisms with high ribosomal production are low in biomass C: P and N: P ratios[14]. Therefore, the *rrn* copy number can explain the growth rate hypothesis[16] since different *rrn* copy numbers vary in biological stoichiometry, affecting bacterial resource demand and growth potential.

Although high *rrn* copy numbers are associated with the abundant biosphere in coastal sediment and the rare biosphere in the free-living fraction of ocean water (Fig. 1b), there is a consistent, positive correlation between the community-level *rrn* copy number and nutrient availability (Supplementary Table 2 and S5). The community-level *rrn* copy number in coastal sediment is $2.74 \pm 0.06$, higher than those of soil and plant-associated microbial community (2.2, based on the Earth Microbiome Project) but lower than animal-associated samples (3.4)[33]. All of those numbers are higher than that in ocean water ($1.54 \pm 0.01$, Fig. 1c) and consistent with their nutrient levels, raising an interesting possibility of using the community-level *rrn* copy number to predict biological nutrient availability in the

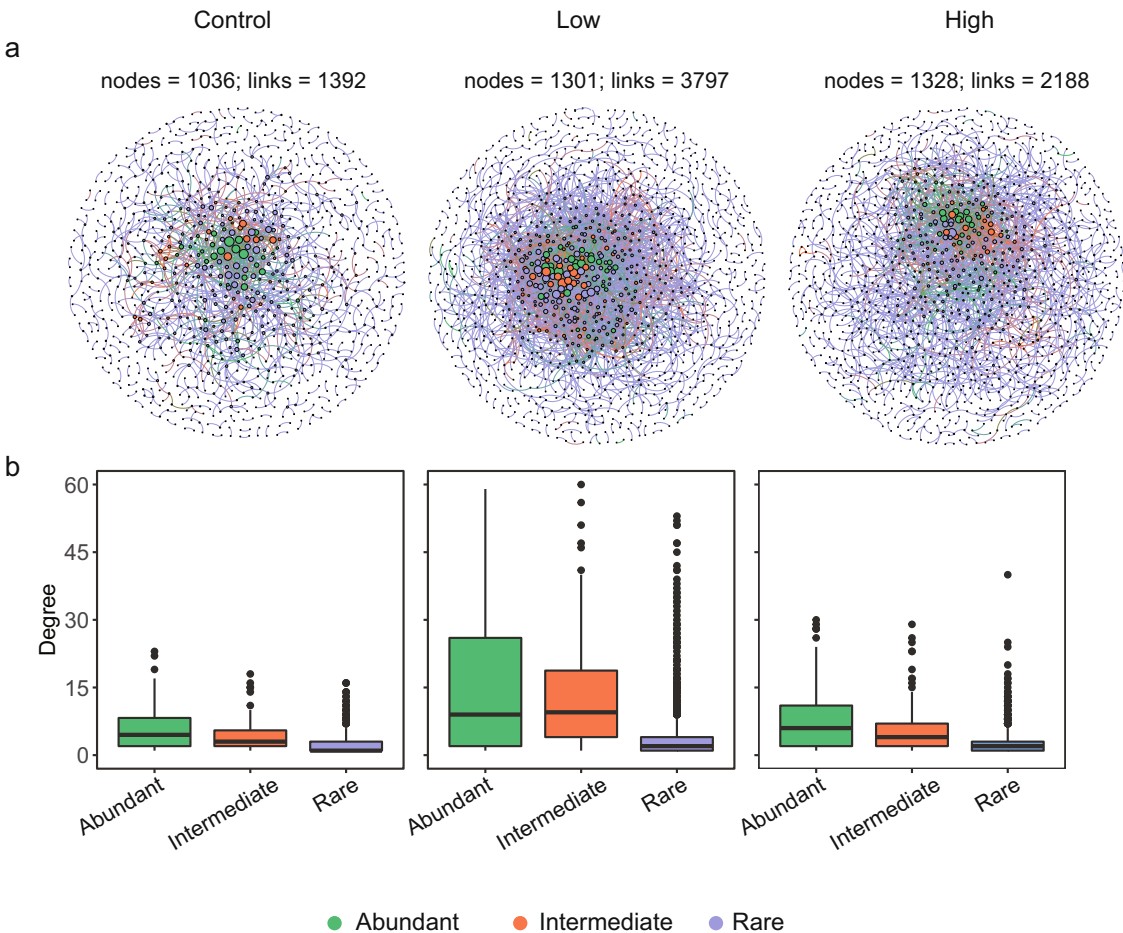

**Fig. 4 The effect of nutrient supply on bacterial network associations. a** The networks were derived from time-series samples (0, 3, 7, 14, 28 days) collected from each of the three nutrient manipulation scenarios: none supply (control), 5 mg L$^{-1}$ NH$_4^+$-N, and 0.5 mg L$^{-1}$ PO$_4^{3-}$-P supply (low), and 50 mg L$^{-1}$ NH$_4^+$-N and 5.0 mg L$^{-1}$ PO$_4^{3-}$-P supply (high). The node color indicates the abundance category of the OTUs, and the node size is proportional to its degree in the network. **b** The boxplots indicate the node degree of OTUs belonging to different abundance categories, where hinges indicate the 25th, 50th, and 75th percentiles of node degree, whiskers indicate 1.5 × interquartile ranges, and dots indicate values beyond those ranges. The sample sizes (i.e., the number of independent OTUs) of the boxplots are as follows: $n = 60$ for abundant, $n = 75$ for intermediate, and $n = 901$ for rare in control; $n = 77$ for abundant, $n = 94$ for intermediate, and $n = 1130$ for rare in low supply; $n = 75$ for abundant, $n = 87$ for intermediate, and $n = 1166$ for rare in high supply. Source data are provided in the Source Data File.

environment in ecological modeling. Nevertheless, disparate phylogenetic lineages with distinct life-history strategies may have identical *rrn* copy numbers, so the impact of phylogenetic structure should be considered when linking community-level *rrn* copy numbers to nutrient availability.

(2) Complex ecological interaction gives rise to the highly dynamic self-organization of microbial communities, termed as an evolutionary game[21]. Competition for scarce food and limited space to survive is common among members in natural bacterial communities[19]. The hunger game strategy prevails, tending to increase species cooperating or competing depending on food availability. Limited nutrient supply reduces secondary metabolite excretion, which contributes to food scarcity of community members utilizing secondary metabolites[34]. Cooperation can result in higher productivity because mutually beneficial species may engage in labor division and exchange essential metabolites, enabling full utilization of nutrients[34,35]. In contrast, competition among members can check and balance each other, leading to a stabilizing effect on the community.

Copiotrophic bacteria outcompete the oligotrophic ones when nutrients are sufficient and vice versa. Unexpectedly, network associations among abundant taxa were predominantly positive in both sediment and ocean water samples (Fig. 2b and Supplementary Table 9), likely due to an equilibrium of stable co-existence and niche partitioning of dominant species arising from competitive exclusion. Low cell density in the oligotrophic ocean water decreased encounter frequency, preventing ecological interactions[19]. Nevertheless, metabolite exchange is especially important for species with small and simple genomes such as SAR11 because metabolic outsourcing is required due to genome reduction[13]. Cooperative growth is preferred under nutrient scarcity (Fig. 2b and Supplementary Table 9), acting as a plausible mechanism to mitigate diversity loss and enhance stability[36,37]. When more nutrient is available, active bacterial growth could cause an intensive modification of the environment by harmful metabolites that inhibited the growth of competitors in the community[21]. The "selfishness" of the winning species impedes species co-existence and thus reduces biodiversity. Active growth may also lead to competition for other essential nutrients, counterbalancing the biotic harshness produced by strong competitors. Nutrient selection can be reduced with consumption, and the metabolites produced by initial winners can open new niches for others, in turn providing new metabolites that support the growth of wider populations[38],

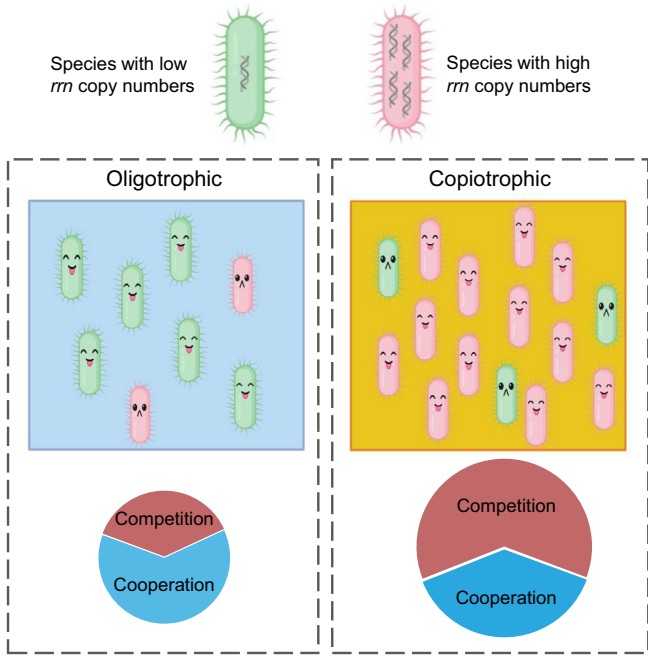

**Fig. 5 The conceptual diagram of the "hunger games" hypothesis. The species abundance in natural bacterial communities is collectively determined by the genomic trait of *rrn* copy number, ecological interaction, and nutrient availability.** Nutrient-scarce environments favor species with low *rrn* copy numbers, while nutrient-rich environment favors species with high *rrn* copy numbers. In nutrient-scarce environments, cooperation, e.g., mutually benefit from metabolite exchange, is promoted to alleviate resource limitation. By contrast, there are more ecological interactions among species in nutrient-rich environments, in which competition plays a larger role.

leading to a recovery of community diversity (Supplementary Fig. 3). Thus, nutrient supply and metabolite excretion simultaneously push the wheel of community succession through different paths. However, the effect of nutrient availability on community stability is uncertain since lower biodiversity reduces stability[39], but a higher negative network association by nutrient amendment observed in both natural and microcosm communities (Fig. 2b and Supplementary Table 9) might enhance stability[40].

However, one should bear in mind that inferring ecological interactions from association networks is feasible only if ecological interactions affect species abundance consistently and properly enough to be detected by adequate statistical methods[41]. To date, it remains intractable how to examine actual ecological interaction in natural environments since all lab experiments are limited to synthetic or simple engineered communities[21,42]. Therefore, association network analysis remains a major tool to infer potential ecological interactions. By focusing on the aggregate pattern of network properties and comparison between networks, we could reduce the impact of false positives and negatives caused by high noise typical in large datasets.

In summary, by comparing the bacterial communities in coastal sediment and the free-living fraction of ocean water, we provide evidence that microbial species abundance profiles can be explained by nutrient availability, the *rrn* copy number, and ecological networks. Our results highlight important roles of bacterial life-history strategy and membership in determining their reproductive success in the natural environment, revealing a fundamental coupling between environmental selection and biotic factors in shaping bacterial communities.

## Methods

**Coastal sediment sampling in Hangzhou Bay, China.** In May 2016, we selected 24 sampling sites in Hangzhou Bay, China, including 9 in the center of Hangzhou Bay, 9 in the wastewater receiving area near Jiaxing City on its north bank, and 6 in the wastewater receiving area near Shangyu City on its south bank (Supplementary Fig. 5). At each site, the surface sediment (0–5 cm) and its overlying seawater were collected in triplicate within a 10 m × 10 m area, resulting in a total of 72 samples (3 replicates × 24 sampling sites). The sediment samples were collected using a stainless-steel grab sampler (Van Veen, Hydro-Bios Apparatebau GmbH, Kiel, Germany) and packed in airtight sterile polypropylene bags. The seawater samples were collected using a Niskin water sampler and stocked in sterile polypropylene bottles. All samples were kept frozen at −20 °C on board. After being transported to the laboratory, the seawater and sediment samples were assayed immediately for physicochemical factors, and the sediment samples were stored at −80 °C for DNA extraction.

**Measurements of seawater and sediment physicochemical factors.** Seawater salinity, pH, and DO were measured in situ. Seawater chemical oxygen demand (COD) was determined by alkalescent permanganate titration. The concentrations of ammonia ($NH_4^+$), nitrite ($NO_2^-$), nitrate ($NO_3^-$), and total phosphorus (TP) in seawater were measured using a spectrophotometer (UV 2401PC, Shimadzu, Kyoto, Japan), by which the detection limits were 0.02 mg/L for $NH_4^+$, 0.01 mg/L for $NO_2^-$, 0.05 mg/L for $NO_3^-$, and 0.0125 mg/L for TP. Sediment organic matter (OM) content was determined by the potassium dichromate oxidation heating method. Sediment total phosphorus (TP) was analyzed by the Mo-Sb antiluminosity method, and total nitrogen (TN) was analyzed by the semi-micro Kjedahl method. The sediment pore-water was extracted by adding 50 mL of 1 M KCl to 10 g sediment, shaked for 1 h, and filtered through 0.45 µm filter. The pore-water dissolved $NH_4^+$, $NO_2^-$, and $NO_3^-$ contents were measured spectrophotometrically.

**Data collection from other studies.** We collected published bacterial 16S rRNA sequencing data on coastal sediment globally (Fig. 1a and Supplementary Data 1), including (i) 4 coastal locations from the Earth Microbiome Project data sets: the Mission Bay in the USA (20 samples), the Gulf of Mexico in the USA (6 samples), the Plymouth Harbor in the UK (65 samples), and the Dois Rios Estuary in Brazil (9 samples); (ii) 3 locations in Mediterranean[43], i.e., Heraklion in Greece (4 samples), Cagliari in Italy (4 samples), and EI Kantaoui in Tunisia (3 samples); and (iii) 4 locations on the coastal Sydney[44]: Narrabeen (15 samples), Dee Why (15 samples), Curl Curl (15 samples), and Manly (15 samples).

Raw sequencing data were downloaded from the NCBI SRA database and processed separately using the same workflow to avoid technical bias (e.g., PCR primer and sequencing platform) from different projects. The sediment physicochemical properties were available only for samples collected from the coastal Mediterranean and coastal Australia. Detailed information about the collected data is shown in Supplementary Data 1.

We also explored the bacterial communities in the global ocean water using metagenomic data generated by the *Tara* Oceans Project, in which free-living prokaryotes with size fractions of 0.22–1.6 µm or 0.22–3 µm were captured[45]. The annotated OTU count table and corresponding representative sequences for a total of 139 prokaryote-enriched samples and the metadata were downloaded from the companion website http://ocean-microbiome.embl.de/companion.html.

**Microcosm experiment.** Seawater-sediment microcosms were constructed to explore the impact of nutrients (N and P) availability on bacterial communities. In May 2017, sediment for microcosm construction was collected from Hangzhou Bay (30.43 N, 121.04 E). Synthetic seawater was spiked with $NH_4Cl$ and $KH_2PO_4$ to manipulate the nutrient availability. As the major nutrient source for coastal water, the effluent of local wastewater treatment plants generally contained 5 mg L$^{-1}$ $NH_4^+$-N and 0.5 mg L$^{-1}$ $PO_4^{3-}$-P[28]. Accordingly, we added 5 mg L$^{-1}$ $NH_4^+$-N and 0.5 mg L$^{-1}$ $PO_4^{3-}$-P as low nutrient supply, and 50 mg L$^{-1}$ $NH_4^+$-N and 5.0 mg L$^{-1}$ $PO_4^{3-}$-P as high nutrient supply. Details about sediment preparation, seawater-sediment microcosm setup, and sampling were described previously[28]. In brief, 25 g sediment was submerged with 50 mL of the manipulated seawater in a 100 mL Erlenmeyer flask. The sediment layer in the microcosm was thin (< 3 cm) so that the added nutrients could be easily accessed by sediment microbes. The microcosms were incubated aerobically at 25 °C in the dark for 28 days. Triplicated microcosms were sacrificed for sampling and microbial community analysis when incubated for 3, 7, 14, and 28 days.

**DNA extraction, bacterial 16S rRNA gene sequencing, and raw data processing.** For the sediment samples collected from Hangzhou Bay and microcosms, total DNA was extracted from 0.25 g of sediment per sample using PowerSoil DNA Isolate Kit (MoBio Laboratories, Carlsbad, CA, USA) according to the manufacture's protocols. The DNA quality, including integrity, purity, and concentration, was checked by 1% agarose and Nanodrop spectrophotometer (Nanodrop Technologies Inc., Wilmington, DE, USA). The DNA samples for PCR amplification were diluted to 2 ng/µL. The primer pair of 338F (5′-ACTCCTACGGG AGGCAGCA-3′) and 806R (5′- GGACTACHVGGGTWTCTAAT-3′)[46–48] was

used to amplify the V3-V4 region of the bacterial 16S rRNA gene, known to cover a broad taxonomic range and have a high resolution for coastal marine bacteria[49]. PCR reactions were performed in triplicate using BioRad S1000 (Bio-Rad Laboratory, Hercules, CA, USA). The thermocycling was: 5 min at 94 °C for initialization; 30 cycles of 30 s denaturation at 94 °C, 30 s annealing at 52 °C, and 30 s extension at 72 °C; followed by 10 min final elongation at 72 °C. Triplicate PCR products were mixed and purified with ENZA Gel Extraction Kit (Omega Bio-Tek Inc., Norcross, GA, USA), and the sequencing library was generated using NEB-Next® Ultra™ DNA Library Prep Kit for Illumina (New England Biolabs, Ipswich, MA, USA) following manufacture's recommendations. At last, the library was sequenced using an Illumina HiSeq 2500 platform (MAGIGENE Co., Ltd., Guangzhou, Guangdong Province, China) to generate 250 bp paired-end reads. Raw sequencing reads were quality controlled using the Trimmomatic (V0.33, http://www.usadellab.org/cms/?page=trimmomatic), and then merged by FLASH (V1.2.11, https://ccb.jhu.edu/software/FLASH/). Operational taxonomic units (OTUs) were clustered at 97% nucleotide identity by UPARSE (V11)[50]. A taxonomic assignment was conducted using the RDP classifier with a confidence cutoff of 50%.

**Estimation of bacterial *rrn* copy number**. The *rrn* copy numbers for bacterial OTUs were estimated based on the *rrn*DB database[27] (version 5.4, https://rrndb.umms.med.umich.edu/). Each OTU was matched with the database starting from the lowest rank. For OTUs with available child taxon matches, the mean *rrn* copy number of all the child taxa was used, otherwise higher rank matches were searched, and the mean *rrn* copy number of the parent taxa for that OTU was assigned. For each sample, the community-level *rrn* copy number, a community level aggregate trait value[24], was calculated as the mean of estimated *rrn* copy number, weighted by the relative abundance for each OTU using Eq. (1) as follows:

$$\text{community} - \text{level } rrn \text{ copy number} = \frac{\sum_{i=1}^{N} S_i}{\sum_{i=1}^{N} \frac{S_i}{n_i}} \qquad (1)$$

where $N$ is the number of OTUs in a sample, $S_i$ is the sequence abundance of OTU$i$, and $n_i$ is the estimated *rrn* copy number of OTU$i$.

**Statistical analyses**. All the statistical analysis was performed based on the R version 3.6.1 (http://www.r-project.org). The differences in OTUs' *rrn* copy number or community-level *rrn* copy number were tested by ANOVA followed by LSD test for multiple comparisons, using the function *LSD.test* in the "agricolae" package. The phylogenetic distance of OTUs in each sample was estimated by the weighted mean pairwise distance (MPD) across a phylogenetic tree[51], using the function *mpd* in the "picante" package. The dissimilarity in bacterial community compositions was tested by non-parametric multivariate analysis using the function *adonis* in the "vegan" package. Pearson's correlations between community-level *rrn* copy number and environmental variables were calculated using the function *Cor.test* in the "stats" package. The linkages between community-level *rrn* copy number and nutrients or community phylogenetic structure were tested by partial Mantel tests using the function *mantel.partial* in the "vegan" package. Euclidean distance was calculated to reveal differences in nutrient availability, and weighted Unifrac distance was calculated to reveal phylogenetic dissimilarity.

**Network construction**. The bacterial association networks were constructed based on the OTUs' relative abundance datasets. OTUs detected in less than 30% sample were discarded to improve the statistical power in correlation calculation while minimizing the OTU loss. The OTUs correlation matrix was calculated using Spearman's rank-based correlation. The appropriate correlation coefficient cutoff for defining the network was determined automatically by a Random Matrix Theory (RMT)-based approach[29]. The network modules were detected by fast greedy modularity optimization. Also, random networks corresponding to each empirical network were constructed by keeping the numbers of nodes and links constant and rewiring the nodes. The topological indexes for the empirical and random networks were calculated as described previously[29,52] using the network analysis pipeline at http://ieg4.rccc.ou.edu/mena/.

**Reporting summary**. Further information on research design is available in the Nature Research Reporting Summary linked to this article.

## Data availability
The raw sequence reads of 16S rRNA gene amplicons for Hangzhou Bay and the microcosm study have been deposited to the National Center for Biotechnology Information (NCBI) Sequence Read Archive (SRA), under BioProject ID PRJNA662822 and PRJNA496525, respectively. The accession numbers for coastal bacterial 16S rRNA gene sequencing data collected from other studies are provided in Supplementary Data 1. The global ocean microbiome data are available in the companion website (http://ocean-microbiome.embl.de/companion.html). The database used for *rrn* copy number estimation is available online (https://rrndb.umms.med.umich.edu/). Source data are provided with this paper.

## Code availability
The R script for classifying the abundant, intermediate, and rare taxa is publicly available on GitHub at https://github.com/TianjiaoDai/HungerGames or under Zenodo at https://doi.org/10.5281/zenodo.5553560.

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

## Acknowledgements

We sincerely thank the support from Dr. Meng Li and Dr. Cuijing Zhang from Shenzhen University in data collection. This work was supported by the China National Key R&D Program (Grant No. 2019YFC1806204 to Y.Y.), the National Natural Science Foundation of China (Grant No. 41825016 to Y.Y., 51938001 to D.W., & 42007291 to T.D.), the China Postdoctoral Science Foundation (No. 2020M670352 to T.D.), the "Shuimu Tsinghua Scholar" program to T.D., and the Hainan Key Laboratory of Tropical Marine Biotechnology Open Fund (No. LTMB201903 to T.D.).

## Author contributions

The original concept was conceived by Y.Y., D.W., and T.D. The field work in Hangzhou Bay was developed by D.W. and T.D. Data from other studies were collected by T.D. with the help from S.L., Y.S., and J.L. The microcosm experiment was designed and carried out by T.D. Data analyses were performed by T.D. with the help from L.W. and X.G. The manuscript was written by T.D., Y.Y., and D.W. with the help from C.B. and J.Z.

## Competing interests

The authors declare no competing interests.
