## [Peer Review File · Nature Communications]

Reviewer comments, first round –

Reviewer #1 (Remarks to the Author):

In this manuscript, Dai et al explored the relationship of nutrient level and community ribosomal rRNA gene (rrn) copy number (as a proxy of microbial growth rate) in both coastal sediment and open ocean samples. They found discrepant patterns of rrn copy numbers in the abundant and rare biosphere of the two distinct ecosystems, suggesting the differential partition of microbial growth potential could be governed by the engrained environmental nutrient supply. Network analysis revealed an increased proportion of negative associations among taxa in coastal sediment microbial communities than in the open ocean communities. Microcosm nutrient addition experiments showed that high nutrient supply correlated with high community rrn copy number. Thus, the authors proposed a “hunger games” hypothesis to depict the nutrient-based control of community growth partition and microbial interaction in the two ecosystems.

This is an interesting manuscript focusing on understanding the relationship of rrn copy number and nutrient availability in natural microbial communities. The topic has great ecological and evolutionary relevance to microbiome studies, the results are well interpreted and discussed, and the manuscript is well written. However, as currently submitted, there are still several concerns preventing it to be considered for publication.

First, it's worth mentioning that rrn copy numbers are phylogenetically conserved on genus and species level (Větrovský et al, 2013, etc). Coastal sediments and open ocean are two evidently different environments from many angles, so it's not surprising that coastal sediments enrich fast-growing taxa and oligotrophic ocean waters enrich slow-growing taxa. Thus, the comparison is confounded by the underlying phylogenetic structure of abundant taxa in each environment. For instance, in the open ocean, SAR11 and picocyanobacteria are the most dominant taxa, they're all oligotrophs with a single copy rrn gene, while one can get multiple abundant OTUs of different ecotypes from different oceanic regions or water columns. Besides, the phylogenetic structure differences can be also scaled up by weighted OTU abundances as did in this study.

Second, the comparison is biased due to size fractionation of open ocean samples. The prokaryotic size fraction of TaraOceans samples only includes microbes with 0.2-3 μm , these are mostly free-living microbes dominating by oligotrophs. In fact, microbial communities associated with particles have a quite different picture where copiotrophs such as Alteromonadales, Oceanospirillales, Flavobacteriales and Rhodospirillales, etc., are more abundant, which have higher growth rates and rrn copy numbers than free-living microbes, and are also commonly found in coastal sediments as shown in this study.

Third, the experimental design of the microcosm study is confusing. I suppose the addition of inorganic nutrients stimulated the growth of primary producers, and the deposition of which further fed the copiotrophic microbes in the sediments. If this is true, then the community changes are due to particle deposition, which is consistent with the particle-associated communities in the open ocean water column. Please clarify if this is not the case.

Lastly, I enjoyed the network analysis part, but I think a more in-depth discussion will be necessary in the scope of competition and cross-feeding in different ecosystems.

Minor Comments:

L225: a high rrn copy number does not necessarily mean high species abundance

L375: please confirm the equation is correct, the denominator is canceled out in this case

Reviewer #2 (Remarks to the Author):

The study by Dai et al., explores the drivers of microbial community dynamics and species abundance in oligotrophic vs. copiotrophic environments. The authors hypothesize that nutrient availability, rrna copy numbers, and microbial networks (competition vs cooperation) play a key role in determining species abundance in the environment. They collected data from 16S rRNA

surveys conducted in coastal sediments and from publicly available Tara ocean data. The authors made additional efforts to examine patterns in culture experiments by manipulating the concentrations of inorganic nitrogen and phosphate and by tracking the microbial community network and rRNA copy numbers.

The manuscript is well written and easy to follow. I have some additional comments below, which are relatively minor in nature. In addition, I find the data analysis to be robust, but have an additional comment regarding how much of the Tara ocean data is skewed by one particular organism, and whether this presents an exception to the rule established by the authors. Nevertheless, I think the proposed "hunger games" mechanism will be of broad interest to the community of microbial ecology.

Specific comments:

Line 30: I understand what the authors mean, but I found this the statement "but increased in coastal waters" slightly vague, or in other words the pattern exhibited the opposite trend to coastal sediments?

Lines 31-32. Does "More negative network associations" translate to fewer microbial networks were observed? For the abstract it may be especially important to clarify this, or describe this in plain language.

Line 32-33: "To verify the effects of nutrient availability..." on rRNA copy numbers or related to network associations? Please clarify.

Line 40: What is meant by "the proportion of negative associations". Please clarify.

Line 52-53: "may undergo conditionally abundance changes" Not sure what you mean here. Please clarify.

Lines 58-59. Does this sentence not require a reference?

Line 85-89. Fairly complicated sentence, consider making these two sentences.

Lines 102-104. With respect to the classification of the abundant, intermediate and rare biosphere indicated in figure 1 caption, is this classification system new to this study, or is this a common cutoff used. A reference here would be helpful, otherwise the cutoff value seems rather arbitrary.

Lines 109-111. The authors note that there were significant differences in rRNA copy numbers among the three biospheres, please consider adding this information to Fig. 1b or Table S2 to indicate the trends that are statistically significant.

Lines 121-122. Given that SAR11 makes up the majority of the Tara ocean OTUs, and has only one rRNA, does this not skew the ocean data to one side. In other words, essentially the ocean data pertaining to the abundant biosphere in Fig. 1b is driven largely by one organism, therefore how robust is this pattern.

Line 130: The average of 1.54 rRNA copies in the global ocean is, however, driven largely by one omnipresent organism, Sar11. I presume that the Tara ocean is largely based on surface water data, hence it would be interesting if this pattern still holds in deeper ocean water samples, where Sar11 is less abundant.

Lines 132-136. Again, the dominance of Sar11 in the abundant category further explains why peak diversity was observed in the intermediate biosphere.

Lines 223-226. "pinpointing the rRNA copy number as the biomarker to predict species abundance profile" Isn't this an overgeneralization, for example, the abundant Sar11 clade is omnipresent in the marine realm, but has one rRNA copy (fewer than rare biosphere species). Hence, I find this idea somewhat difficult to reconcile, perhaps the authors could speak to this point.

Lines 236-237: But the low value reported in the Tara ocean is driven by one ubiquitous clade, otherwise the intermediate and rare rrn copy numbers are similar to the values reported in sediments, if I see this correctly from Fig. 1b. This is an important caveat worth mentioning.

Line 257. Drop the word "which".

Line 303. The authors state "standard methods", could they be a bit more specific. Are they referring to photometric techniques? Moreover, it's also good practice to report the nutrient detection limits for the measured substrates.

Lines 305-306. The reference is mentioned, but briefly add some additional detail regarding the extraction technique and instrument used.

Lines 334-335. Do the authors have a specific reasoning behind why they used these concentrations – please clarify.

Lines 349-351. Seems to be the wrong reference (ref 45) for the primer set used? Please double check and provide correct reference. The reference is referring to a cancer research paper with no mention of primer 338F. Could the authors also comment on primer bias, how good is the primer coverage, and is there a possibility that they favour abundant organisms over rare organisms possibly skewing the data. In other words, are these primers effective at targeting the full diversity associated with the microbial biosphere.

Lines 394-395. "OTUs detected in less than 30% of samples were removed to minimized false-positive results." Is there a specific reason for using 30%, otherwise it comes across as being rather arbitrary – please clarify.

A. Responses to Reviewer #1 (Remarks to the Author)

A1.

In this manuscript, Dai et al explored the relationship of nutrient level and community ribosomal rRNA gene (rrn) copy number (as a proxy of microbial growth rate) in both coastal sediment and open ocean samples. They found discrepant patterns of rrn copy numbers in the abundant and rare biosphere of the two distinct ecosystems, suggesting the differential partition of microbial growth potential could be governed by the engrained environmental nutrient supply. Network analysis revealed an increased proportion of negative associations among taxa in coastal sediment microbial communities than in the open ocean communities. Microcosm nutrient addition experiments showed that high nutrient supply correlated with high community rrn copy number. Thus, the authors proposed a “hunger games” hypothesis to depict the nutrient-based control of community growth partition and microbial interaction in the two ecosystems.

This is an interesting manuscript focusing on understanding the relationship of rrn copy number and nutrient availability in natural microbial communities. The topic has great ecological and evolutionary relevance to microbiome studies, the results are well interpreted and discussed, and the manuscript is well written.

Response: Thank you very much for your kind encouragement!

However, as currently submitted, there are still several concerns preventing it to be considered for publication.

First, it's worth mentioning that rrn copy numbers are phylogenetically conserved on genus and species level (Větrovský et al, 2013, etc). Coastal sediments and open ocean are two evidently different environments from many angles, so it's not surprising that coastal sediments enrich fast-growing taxa and oligotrophic ocean waters enrich slow-growing taxa. Thus, the comparison is confounded by the underlying phylogenetic structure of abundant

taxa in each environment. For instance, in the open ocean, SAR11 and picocyanobacteria are the most dominant taxa, they're all oligotrophs with a single copy rrn gene, while one can get multiple abundant OTUs of different ecotypes from different oceanic regions or water columns. Besides, the phylogenetic structure differences can be also scaled up by weighted OTU abundances as did in this study.

Response: We agree that the *rrn* copy number is phylogenetically conserved. Therefore, we have followed the suggestion to mention it in lines 85-87 “The ribosomal RNA gene operon (*rrn*) copy number in bacterial genomes is a phylogenetically conserved trait on genus and species levels²², which predict the growth rate and nutrient utilization efficiency for individual organisms well^{10,11,23}.”

We have calculated the abundance weighted community-level *rrn* copy numbers on both the genus and OTU levels and found that they are similar, which is shown in the Table below. When abundance unweighted community-level *rrn* copy numbers were calculated, the values were still significantly higher in coastal sediments than ocean water, despite the differences were smaller compared to those of the abundance weighted ones. Those results demonstrate that analyzing bacterial communities on the genus or OTU level, with or without weighted abundance, will not change our conclusion. In addition, the abundant taxa in the same environment tend to have similar *rrn* copy numbers (e.g., SAR11 and picocyanobacteria in the oligotrophic open ocean), though the *rrn* copy number is not distinguishable among ecotypes. Those findings have confirmed the selection of high *rrn* copy number under copiotrophic environments and low *rrn* copy number under oligotrophic environments as well. We have added the data of abundance unweighted community-level *rrn* copy numbers to Table S2. We have also added a sentence in lines 135-138 “The abundance unweighted community-level *rrn* copy numbers in ocean water were also lower than those in sediments (Table S2), suggesting that ignoring taxon abundance will not affect our results.”

Comparison of the community-level average *rrn* copy numbers

Location	OTU level		Genus level	
	abundance weighted	abundance unweighted	abundance weighted	abundance unweighted
Coastal sediment				
Mission Bay	3.52±0.25 a	2.89±0.09 a	3.46±0.18 a	2.99±0.08 a
Hangzhou Bay	3.18±0.17 ab	2.18±0.01 d	3.23±0.16 ab	2.22±0.01 d
the Gulf of Mexico	2.79±0.06 bc	2.51±0.01 b	2.82±0.06 bc	2.61±0.01 b
Plymouth Harbor	2.59±0.02 c	2.45±0.02 b	2.76±0.03 cd	2.59±0.02 b
Coastal Mediterranean	2.38±0.01 c	2.34±0.01 c	2.49±0.02 c	2.45±0.01 c
Coastal Sydney	2.24±0.01 c	2.24±0.01 d	2.41±0.01 c	2.39±0.01 c
Dois Rios Estuary	2.16±0.16 c	2.26±0.07 cd	2.30±0.16 c	2.35±0.08 c
Ocean water				
the Tara Oceans	1.54±0.01 d	2.18±0.01 d	1.73±0.02 d	2.05±0.02 e

Lowercase letters indicate significant differences based on the LSD test after ANOVA.

A2.

Second, the comparison is biased due to size fractionation of open ocean samples. The prokaryotic size fraction of TaraOceans samples only includes microbes with 0.2-3 μm, these are mostly free-living microbes dominating by oligotrophs. In fact, microbial communities associated with particles have a quite different picture where copiotrophs such as Alteromonadales, Oceanospirillales, Flavobacteriales and Rhodospirillales, etc., are more abundant, which have higher growth rates and rrn copy numbers than free-living microbes, and are also commonly found in coastal sediments as shown in this study.

Response: Yes, the prokaryotic size fraction of Tara Oceans samples mainly includes free-living microbes¹. Due to the lack of particle-associated microbial profile in the Tara Oceans samples, we cannot assess the particle-associated fraction. We have explicitly indicated it in lines 342-344 “We also explored the bacterial communities in the global ocean water using metagenomic data generated by the Tara Oceans Project, in which free-living prokaryotes with size fractions of 0.22-1.6 μm or 0.22-3 μm were captured⁴⁴.” We respectfully ask for your permission to include your comment in the revised manuscript, shown in lines 225-229

“However, the particle-associated microbial communities have a quite different picture where copiotrophs such as *Alteromonadales*, *Oceanospirillales*, *Flavobacteriales*, and *Rhodospirillales*, are more abundant, which have higher growth rates and *rrn* copy numbers than free-living microbes⁹ and are also commonly found in coastal sediments as shown in this study.”

A3.

Third, the experimental design of the microcosm study is confusing. I suppose the addition of inorganic nutrients stimulated the growth of primary producers, and the deposition of which further fed the copiotrophic microbes in the sediments. If this is true, then the community changes are due to particle deposition, which is consistent with the particle-associated communities in the open ocean water column. Please clarify if this is not the case.

Response: In our microcosm experiments, the microcosms were incubated in the dark, so the growth of photoautotrophic primary producers should be negligible. As a result, feeding the copiotrophic microbes in the sediments by particle deposition of primary producers should be minor, if any. Rather, the sediment in the microcosm was thin (<3 cm), so added nutrients could easily be accessed by microbes in the sediment. We have clarified it in lines 360-362 “The sediment layer in the microcosm was thin (<3 cm) so that the added nutrients could be easily accessed by sediment microbes. The microcosms were incubated aerobically at 25 °C in the dark for 28 days.”

A4.

Lastly, I enjoyed the network analysis part, but I think a more in-depth discussion will be necessary in the scope of competition and cross-feeding in different ecosystems.

Response: We have followed the suggestion to add more discussions, as shown in lines 258-268 “Copiotrophic bacteria outcompete the oligotrophic ones when nutrients are sufficient and vice versa. Unexpectedly, network associations among abundant taxa were

predominantly positive in both sediment and ocean water samples (Figure 2b & Table S9), likely due to an equilibrium of stable co-existence and niche partition of dominant species arising from competitive exclusion. Low cell density in the oligotrophic ocean water decreased encounter frequency, preventing ecological interactions¹⁹. Nevertheless, metabolite exchange is especially important for species with small and simple genomes such as SAR11 because metabolic outsourcing is required due to genome reduction¹³. Cooperative growth is preferred under nutrient scarcity (Figure 2b and Table S9), acting as a plausible mechanism to mitigate diversity loss and enhance stability^{35,36}.” and lines 271-273 “Active growth may also lead to competition for other essential nutrients, counterbalancing the biotic harshness produced by strong competitors.”

Minor Comments:

A5.

L225: a high rrn copy number does not necessarily mean high species abundance

Response: Yes, we agree. So we have revised the sentence in lines 231-233: “This finding advances the life-history strategy hypothesis for rare and abundant biospheres^{3,8} by pinpointing the rrn copy number as a genomic feature related to species reproduction success.”

A6.

L375: please confirm the equation is correct, the denominator is canceled out in this case

Response: We are sorry that we made a mistake by missing brackets in the numerator. We have corrected the equation below:

$$\text{community-level } rrn \text{ copy number} = \frac{\sum_{i=1}^N \left(\frac{S_i}{n_i} \times n_i \right)}{\sum_{i=1}^N \frac{S_i}{n_i}} = \frac{\sum_{i=1}^N S_i}{\sum_{i=1}^N \frac{S_i}{n_i}}$$

where N is the number of OTUs in a sample, S_i is the sequence abundance of OTU $_i$, n_i is the estimated *rrn* copy number of OTU $_i$, and $\frac{S_i}{n_i}$ represents the number of individuals of OTU $_i$.

We have presented this equation in line 401 in the revised manuscript.

B. Responses to Reviewer #2 (Remarks to the Author)

B1.

The study by Dai et al., explores the drivers of microbial community dynamics and species abundance in oligotrophic vs. copiotrophic environments. The authors hypothesize that nutrient availability, rRNA copy numbers, and microbial networks (competition vs cooperation) play a key role in determining species abundance in the environment. They collected data from 16S rRNA surveys conducted in coastal sediments and from publicly available Tara ocean data. The authors made additional efforts to examine patterns in culture experiments by manipulating the concentrations of inorganic nitrogen and phosphate and by tracking the microbial community network and rRNA copy numbers.

The manuscript is well written and easy to follow. I have some additional comments below, which are relatively minor in nature. In addition, I find the data analysis to be robust, but have an additional comment regarding how much of the Tara ocean data is skewed by one particular organism, and whether this presents an exception to the rule established by the authors. Nevertheless, I think the proposed “hunger games” mechanism will be of broad interest to the community of microbial ecology.

Response: Thanks for your encouragement! It is true that Tara ocean data is skewed by the highly abundant SAR11 clade in the surface ocean water, which is also mentioned in your comments B11-15. To test it, we have performed additional analyses. We summarize our findings here, which are also echoed in our responses to B11-15.

When SAR11 was removed from the dataset, the observation that “*rrn* copy number increased from rare to abundant biosphere in ocean water” remains the same. SAR11 abundance decreased but the community-level *rrn* copy number for ocean water increased from the surface (0-200 m) to the deep ocean (200-1000 m). However, the community-level *rrn* copy number in the deep ocean was still much lower than those of the sediment communities. Therefore, removing SAR11 will not affect our findings that (i) low *rrn* copy

number is favored in oligotrophic environments; and (ii) community-level *rrn* copy number could reflect biological nutrient availability.

We have added the results to Figure S1 and S2. We have described the results in lines 120-124 “To assess how SAR11 affects *rrn* copy numbers, we removed all of SAR11 OTUs from the datasets. We showed that the OTUs’ *rrn* copy numbers of the abundant biosphere remained the lowest for ocean water samples (Figure S1), suggesting that our finding remains to be robust when removing SAR11.” and lines 131-135 “The community-level *rrn* copy number in global ocean water was higher in deep ocean than surface ocean water, which could be attributed to lower SAR11 abundance (Figure S2). However, it averaged 1.54 ± 0.01 within a range between 1.39 and 2.53 (Figure 1c), which was still significantly ($P < 0.001$, ANOVA) lower than sediments.”

Specific comments:

B2.

Line 30: I understand what the authors mean, but I found this the statement “but increased in coastal waters” slightly vague, or in other words the pattern exhibited the opposite trend to coastal sediments?

Response: We have clarified it in lines 27-31 as: “The ribosomal RNA gene operon (*rrn*) copy number, a genomic trait related to bacterial growth rate and nutrient demand, decreased from abundant to rare biosphere in coastal sediments but exhibited the opposite trend in ocean waters, which could be unified by positive correlations between community-level *rrn* copy number and nutrients.”

B3.

Lines 31-32. Does “More negative network associations” translate to fewer microbial networks were observed? For the abstract it may be especially important to clarify this, or describe this in plain language.

Response: No, it means more negative correlations of OTUs’ abundances were observed in the network, reflecting a higher possibility of inter-species co-exclusion. We have described it in more plain language in lines 31-33 “Inter-species co-exclusion reflected by negative network associations were more observed in sediment samples than ocean water.”

B4.

Line 32-33: “To verify the effects of nutrient availability...” on rrna copy numbers or related to network associations? Please clarify.

Response: We meant to verify the effects of nutrient availability on both rrn copy numbers and network associations. To clarify it, we have revised the sentences in lines 33-35 “To verify the effects of nutrient availability on rrn copy numbers and network associations, we experimentally manipulated nitrogen and phosphorus supplies in microcosms and found similar results to field observations.”

B5.

Line 40: What is meant by “the proportion of negative associations”. Please clarify.

Response: As indicated in B3, more negative correlations mean a higher possibility of inter-species co-exclusion. We have clarified it in lines 40-42 “low nutrient supply decreases the degree of potential ecological interaction and reduces inter-species co-exclusion, suggesting a game strategy of bacteria in response to food scarcity.”

B6.

Line 52-53: “may undergo conditionally abundance changes” Not sure what you mean here. Please clarify.

Response: Here, we mean that abundances of certain microbial taxa depend on whether the environmental conditions are suitable for growth, thus the abundances are highly variable across our samples. We have revised the sentence in lines 52-53 “As a result, some species are persistent members of the rare or abundant biosphere, and others are variable in abundance³.”

B7.

Lines 58-59. Does this sentence not require a reference?

Response: We have added two references in line 60.

- Roller BRK, Stoddard SF, Schmidt TM. Exploiting rRNA operon copy number to investigate bacterial reproductive strategies. *Nat Microbiol* **1**, (2016).
- Giovannoni SJ. SAR11 Bacteria: The Most Abundant Plankton in the Oceans. *Annual Review of Marine Science* **9**, 231-255 (2017).

B8.

Line 85-89. Fairly complicated sentence, consider making these two sentences.

Response: We have followed the suggestion to split it into two sentences in lines 85-90 “The ribosomal RNA gene operon (*rrn*) copy number in bacterial genomes is a phylogenetically conserved trait on genus and species levels²², which predict the growth rate and nutrient utilization efficiency for individual organisms well^{10, 11, 23}. Accordingly, we examined whether the average *rrn* copy number of community members was positively correlated with environmental nutrient contents, which was verified in natural²⁴ and engineered systems^{25, 26.}”

B9.

Lines 102-104. With respect to the classification of the abundant, intermediate and rare biosphere indicated in figure 1 caption, is this classification system new to this study, or is this a common cutoff used. A reference here would be helpful, otherwise the cutoff value seems rather arbitrary.

Response: No, our classification system is not new. Here, we have adopted commonly used abundance and occurrence frequency cutoffs. So we have added three references in the legend of Figure 1.

- Wu LW, et al. Global diversity and biogeography of bacterial communities in wastewater treatment plants. *Nat Microbiol* **4**, 2579-2579 (2019).
- Ju F, Zhang T. Bacterial assembly and temporal dynamics in activated sludge of a full-scale municipal wastewater treatment plant. *Isme J* **9**, 683-695 (2015).
- Galand PE, Casamayor EO, Kirchman DL, Lovejoy C. Ecology of the rare microbial biosphere of the Arctic Ocean. *Proc Natl Acad Sci USA* **106**, 22427-22432 (2009).

B10.

Lines 109-111. The authors note that there were significant differences in rrn copy numbers among the three biospheres, please consider adding this information to Fig. 1b or Table S2 to indicate the trends that are statistically significant.

Response: We have indicated significant differences among the three biospheres with lowercase letters above the bars in Figure 1b and Table S2, based on the LSD test after ANOVA (adjusted $P < 0.05$). We have revised the figure legend in Figure 1 to indicate it as “In panels b, c, and d, error bars represent the standard errors and lowercase letters above the bars indicate significant differences based on the LSD test after ANOVA (adjusted $P < 0.05$).” We have also revised the footnote of Table S2 as “The rrn copy numbers of OTUs are presented as mean \pm s.e., where lowercase letters in bold indicate significant difference based on LSD test after ANOVA, and data in the brackets indicate the proportion of OTUs in the dataset.”

B11.

Lines 121-122. Given that SAR11 makes up the majority of the Tara ocean OTUs, and has only one *rrn*, does this not skew the ocean data to one side. In other words, essentially the ocean data pertaining to the abundant biosphere in Fig. 1b is driven largely by one organism, therefore how robust is this pattern.

Response: To test the concern raised by the reviewer, we have removed all OTUs affiliated to SAR11 clade from the original dataset and re-classified the rare, intermediate, and abundant biosphere. In the new dataset, the proportion of abundant biosphere OTUs with single *rrn* copy decreased (see below), suggesting the existence of SAR11 has indeed skewed the dataset. However, the *rrn* copy number of the abundant biosphere is still the lowest after SAR11 were removed. Therefore, the pattern that “*rrn* copy number increased from abundant to rare biosphere in ocean water (Fig. 1b)” is unaffected by SAR11. We have added panel b as Figure S1, and added a sentence in lines 120-124 “To assess how SAR11 affects *rrn* copy numbers, we removed all of SAR11 OTUs from the dataset. We showed that the OTUs’ *rrn* copy numbers of the abundant biosphere remained the lowest for ocean water samples (Figure S1), suggesting that our finding remains to be robust when removing SAR11.”

The impact of SAR11 on the “*rrn* copy number - abundance” patterns in the Tara Oceans dataset. **a**, the *rrn* copy number distribution of the OTUs belonging to the abundant biosphere. **b**, OTUs’ *rrn* copy number of the abundant, intermediate, and rare biospheres

B12.

Line 130: The average of 1.54 rrn copies in the global ocean is, however, driven largely by one omnipresent organism, Sar11. I presume that the Tara ocean is largely based on surface water data, hence it would be interesting if this pattern still holds in deeper ocean water samples, where Sar11 is less abundant.

Response: The Tara Ocean dataset included both surface (0-200 m) and deep ocean samples (200-1000 m). SAR11 abundance decreased but the community-level *rrn* copy number for ocean water increased from the surface (0-200 m) to the deep ocean (200-1000 m) (see below). However, the community-level *rrn* copy number in the deep ocean (1.62 ± 0.04) was still lower ($P < 0.001$, ANOVA) than the sediment (2.74 ± 0.06), suggesting the observation that lower community-level *rrn* copy number in ocean water than sediment still holds true for the deep ocean. We have added the figure as Figure S2, and revised the sentence in lines 131-135 “The community-level *rrn* copy number in global ocean water was higher in deep ocean than surface ocean water, which could be attributed to lower SAR11 abundance (Figure S2). However, it averaged 1.54 ± 0.01 within a range between 1.39 and 2.53 (Figure 1c), which was still significantly ($P < 0.001$, ANOVA) lower than sediments.”

Figure S2 The relative abundance of SAR11 and the community community-level *rrn* copy number from the surface to the deep ocean. The dashed lines in **a** and **b** divide the surface ocean (0 - 200 m) and the deep ocean (200 - 1000 m). Adjusted R^2 and P values from linear regression are shown.

B13.

Lines 132-136. Again, the dominance of Sar11 in the abundant category further explains why peak diversity was observed in the intermediate biosphere.

Response: Here, we meant the diversity of the whole community, not the intermediate biosphere. To clarify it, we have revised the sentence in lines 138-142 “By calculating the mean pairwise distance (MPD) of OTUs across the phylogenetic tree, we found that the communities with the highest diversity (i.e., the coastal Mediterranean and coastal Sydney in Figure 1d) did not have the highest or lowest community-level *rrn* copy numbers (Figure 1c), suggesting OTUs in these communities covered a broader phylogenetic lineage.”

B14.

*Lines 223-226. “pinpointing the *rrn* copy number as the biomarker to predict species abundance profile” Isn’t this an overgeneralization, for example, the abundant Sar11 clade is omnipresent in the marine realm, but has one *rrn* copy (fewer than rare biosphere species). Hence, I find this idea somewhat difficult to reconcile, perhaps the authors could speak to this point.*

Response: We agree that it is an overgeneralization. As *rrn* copy number is not a good biomarker of species abundance profile, we have revised the sentence in lines 231-233 “This finding advances the life-history strategy hypothesis for rare and abundant biospheres^{3,8} by pinpointing the *rrn* copy number as a genomic feature related to species reproduction success.”

B15.

*Lines 236-237: But the low value reported in the Tara ocean is driven by one ubiquitous clade, otherwise the intermediate and rare *rrn* copy numbers are similar to the values*

reported in sediments, if I see this correctly from Fig. 1b. This is an important caveat worth mentioning.

Response: When removing the ubiquitous clade SAR11 from the Tara Ocean dataset, the community-level *rrn* copy number in ocean water is 2.12 ± 0.02 , still lower than the coastal sediment (2.74 ± 0.06), suggesting that highly abundant SAR11 did not change the pattern. We have also calculated the abundance unweighted community-level *rrn* copy numbers to minimize the impact of abundant taxa. Lower values in ocean water than coastal sediment was detected (Table S2), suggesting that abundant taxa did not change the pattern. We have added those data in the revised manuscript together with text in lines 135-138 “The abundance unweighted community-level *rrn* copy numbers in ocean water were also lower than those in sediments (Table S2), suggesting that ignoring taxon abundance will not affect our results.”

B16.

Line 257. Drop the word “which”.

Response: We have followed the suggestion to drop it.

B17.

Line 303. The authors state “standard methods”, could they be a bit more specific. Are they referring to photometric techniques? Moreover, it’s also good practice to report the nutrient detection limits for the measured substrates.

Response: Yes, most of the “standard methods” used are photometric techniques. We have followed the suggestion to provide a more specific description and report the nutrient detection limits for the measured substrates in lines 315-323 “Seawater chemical oxygen demand (COD) was determined by alkalescent permanganate titration. The concentrations of ammonia (NH_4^+), nitrite (NO_2^-), nitrate (NO_3^-), and total phosphorus (TP) in seawater were

measured using a spectrophotometer (UV 2401PC, Shimadzu, Kyoto, Japan), by which the detection limits were 0.02 mg/L for NH₄⁺, 0.01 mg/L for NO₂⁻, 0.05 mg/L for NO₃⁻, and 0.0125 mg/L for TP. Sediment organic matter (OM) content was determined by the potassium dichromate oxidation heating method. Sediment total phosphorus (TP) was analyzed by the Mo-Sb antiluminosity method, and total nitrogen (TN) was analyzed by the semi-micro Kjeldahl method.”

B18.

Lines 305-306. The reference is mentioned, but briefly add some additional detail regarding the extraction technique and instrument used.

Response: We added details in lines 323-325 “The sediment pore-water was extracted by adding 50 mL of 1 M KCl to 10 g sediment, shaking for 1 h, and filtering through 0.45 μm filter. The pore-water dissolved NH₄⁺, NO₂⁻, and NO₃⁻ contents were measured spectrophotometrically.”

B19.

Lines 334-335. Do the authors have a specific reasoning behind why they used these concentrations – please clarify.

Response: Yes, we have a specific reason for using these concentrations. We have clarified it in lines 354-357 “As the major nutrient sources for coastal water, the effluent of local wastewater treatment plants generally contained 5 mg L⁻¹ NH₄⁺-N and 0.5 mg L⁻¹ PO₄³⁻-P²⁸. Accordingly, we added 5 mg L⁻¹ NH₄⁺-N and 0.5 mg L⁻¹ PO₄³⁻-P as low nutrient supply, and 50 mg L⁻¹ NH₄⁺-N and 5.0 mg L⁻¹ PO₄³⁻-P as high nutrient supply.”

B20.

Lines 349-351. Seems to be the wrong reference (ref 45) for the primer set used? Please double check and provide correct reference. The reference is referring to a cancer research

paper with no mention of primer 338F. Could the authors also comment on primer bias, how good is the primer coverage, and is there a possibility that they favour abundant organisms over rare organisms possibly skewing the data. In other words, are these primers effective at targeting the full diversity associated with the microbial biosphere.

Response: We have corrected the references as:

- Huse SM, Dethlefsen L, Huber JA, Welch DM, Relman DA, Sogin ML. Exploring Microbial Diversity and Taxonomy Using SSU rRNA Hypervariable Tag Sequencing. *Plos Genet* **4**, e1000255 (2008).
- Caporaso JG, et al. Global patterns of 16S rRNA diversity at a depth of millions of sequences per sample. *P Natl Acad Sci USA* **108**, 4516-4522 (2011).
- Salas-González I, et al. Coordination between microbiota and root endodermis supports plant mineral nutrient homeostasis. *Science* **371**, eabd0695 (2021).

For coastal marine microbial community study, the performance of several primers targeting V3-V4, V4, and V4-V5 hypervariable regions of the 16S rRNA gene have been compared². It showed that V3-V4 regions have a broader taxonomic range and the best resolution but bias against the SAR11 and archaea, while the V4-V5 (515F-926R) and V4 (515F-806R) regions are superior for simultaneous bacterial and archaeal characterization. Considering SAR11 is of low abundance in coastal sediment³, and archaea were not considered in this study, we have decided to target the V3-V4 regions using 338F-806R for coastal sediment bacterial diversity analysis. We have commented on the primer choice in lines 372-376 “The primer pair of 338F (5'- ACTCCTACGGGAGGCAGCA-3') and 806R (5'- GGACTACHVGGGTWTCTAAT-3')^{45,46,47} was used to amplify the V3-V4 region of bacterial 16S rRNA gene, known to cover a broad taxonomic range and have a high resolution for coastal marine bacteria⁴⁸.”

B21.

Lines 394-395. “OTUs detected in less than 30% of samples were removed to minimized false-positive results.” Is there a specific reason for using 30%, otherwise it comes across as being rather arbitrary – please clarify.

Response: Yes, there is a specific reason for using 30%. The reported occurrence frequency thresholds in network analysis ranged widely from 20% to 80%^{4, 5, 6, 7}. When using 40% or higher values to our datasets, associations among rare taxa were not detectable because too many rare taxa were removed. Both 20% and 30% are fine, but a higher threshold value can improve the statistical power of correlation analysis because the degree of freedom is larger. Therefore, we have used 30%. We have clarified it in lines 419-420 “OTUs detected in less than 30% sample were discarded to improve the statistic power of correlation calculation while minimizing the OTU loss.”

References

1. Pesant S, *et al.* Open science resources for the discovery and analysis of Tara Oceans data. *Sci Data* **2**, (2015).
2. Wear EK, Wilbanks EG, Nelson CE, Carlson CA. Primer selection impacts specific population abundances but not community dynamics in a monthly time-series 16S rRNA gene amplicon analysis of coastal marine bacterioplankton. *Environ Microbiol* **20**, 2709-2726 (2018).
3. Klindworth A, *et al.* Evaluation of general 16S ribosomal RNA gene PCR primers for classical and next-generation sequencing-based diversity studies. *Nucleic Acids Res* **41**, (2013).
4. Ju F, Zhang T. Bacterial assembly and temporal dynamics in activated sludge of a full-scale municipal wastewater treatment plant. *Isme J* **9**, 683-695 (2015).
5. Ju F, Li B, Ma LP, Wang YB, Huang DP, Zhang T. Antibiotic resistance genes and human bacterial pathogens: Co-occurrence, removal, and enrichment in municipal sewage sludge digesters. *Water Res* **91**, 1-10 (2016).
6. Yuan MM, *et al.* Climate warming enhances microbial network complexity and stability. *Nat Clim Change* **11**, 343-U100 (2021).
7. Wu LW, *et al.* Long-term successional dynamics of microbial association networks in anaerobic digestion processes. *Water Res* **104**, 1-10 (2016).

Reviewer comments, second round –

Reviewer #1 (Remarks to the Author):

A1: The authors recalculated abundance weighted/unweighted community rrn copy numbers on both genus and OTU levels, and found coastal sediments still have higher rrn copy numbers than oceanic waters. I appreciate the effort they put, while the question is about the bias introduced by underlying phylogenetic structure in each ecosystem rather than the taxonomy ranks used. As the authors pointed out, the difference in rrn copy numbers were less obvious using abundance unweighted calculation, further confirming that phylogenetic structure matters. This can be well demonstrated by the Tara samples with the Hangzhou Bay samples, where a remarkable difference can be observed in the abundance weighted calculation, while the unweighted ones have very similar rrn copy numbers.

A2: Thanks a lot for requesting my permission, I'm happy to share ideas if it helps. As argued by the other reviewer (B1), the presence of abundance oligotrophs indeed skewed the analysis. In deeper water columns (200-1000 m as tested here), where bacterial remineralization of sinking particles happening, the community rrn copy numbers were higher than in the surface waters, suggesting the availability of organic matter is essential. This is similar to the particle-attached fraction of oligotrophic open ocean samples, where fast-growing copiotrophs are enriched, thus higher community rrn copy numbers are expected. I understand compiling particle-attached metagenomic datasets for additional analysis might require too much effort and time, if the authors are not willing to, I would suggest removing all the OTUs that are predicted to have only one rrn copy to avoid this bias.

A3: Thanks a lot for clarifying.

A4: Thanks a lot, I'm satisfied with the current modification.

Reviewer #2 (Remarks to the Author):

In this revision, Dai and co-authors have done a good job at addressing the questions and comments previously raised by myself and the other reviewer. The authors have clarified sections of the abstract and text that were previously vague, and have provided additional info regarding particular methods, which is appreciated. The authors have also convincingly shown that the data and patterns observed are robust, with and without Sar11 in the Tara Ocean dataset analysis, as they indicated in Figure. S1. As mentioned before, this is an impressive and comprehensive dataset and I look forward to seeing this study in its final published version.

A. Response to Reviewer #1 (Remarks to the Author)

A1: *The authors recalculated abundance weighted/unweighted community rrn copy numbers on both genus and OTU levels, and found coastal sediments still have higher rrn copy numbers than oceanic waters. I appreciate the effort they put, while the question is about the bias introduced by underlying phylogenetic structure in each ecosystem rather than the taxonomy ranks used. As the authors pointed out, the difference in rrn copy numbers were less obvious using abundance unweighted calculation, further confirming that phylogenetic structure matters. This can be well demonstrated by the Tara samples with the Hangzhou Bay samples, where a remarkable difference can be observed in the abundance weighted calculation, while the unweighted ones have very similar rrn copy numbers.*

Response: We agree with the comment! Therefore, we have added a sentence in lines 255-257 as “Nevertheless, disparate phylogenetic lineages with distinct life-history strategies may have identical rrn copy numbers, so the impact of phylogenetic structure should be considered when linking community-level rrn copy numbers to nutrient availability.”

A2: *Thanks a lot for requesting my permission, I'm happy to share ideas if it helps. As argued by the other reviewer (B1), the presence of abundance oligotrophs indeed skewed the analysis. In deeper water columns (200-1000 m as tested here), where bacterial remineralization of sinking particles happening, the community rrn copy numbers were higher than in the surface waters, suggesting the availability of organic matter is essential. This is similar to the particle-attached fraction of oligotrophic open ocean samples, where fast-growing copiotrophs are enriched, thus higher community rrn copy numbers are expected. I understand compiling particle-attached metagenomic datasets for additional analysis might require too much effort and time, if the authors are not willing to, I would suggest removing all the OTUs that are predicted to have only one rrn copy to avoid this bias.*

Response: We have followed the suggestion to remove OTUs with a single rrn copy and

re-classify the rare, intermediate, and abundant biosphere. As a result, the abundant biosphere in ocean water was dominated by OTUs with 2 *rrn* copies, and still had significantly ($P < 0.05$) lower *rrn* copy numbers than the intermediate and rare biosphere (see figures below). These results suggested that our finding remains robust when considering potential bias induced by the abundant oligotrophs in ocean water. We have replaced Figure S1 with panel **b** and revised the sentences in lines 121-126 as “To assess how SAR11 and other OTUs with a single *rrn* copy affect *rrn* copy numbers, we removed those OTUs from the datasets. The OTUs’ *rrn* copy numbers of the abundant biosphere remained lower than the intermediate and rare biospheres (Figure S1), confirming the enrichment of taxa with fewer *rrn* copies in ocean water.” The relative sequence abundance of OTUs with a single *rrn* copy in ocean water averaged 11.67% (in the range of 0.91% to 31.12%). Since the presence of these OTUs is a natural phenomenon, we have decided to keep them when calculating the community-level *rrn* copy number. Otherwise, the original community structure is biased.

We agree with your comment that the higher organic matter availability leads to the enrichment of copiotrophs in the particle-attached fraction of ocean water, and in the deep ocean water. So we have explicitly pointed it out in lines 230-236 as “However, the particle-associated microbial communities have a quite different picture where copiotrophs such as *Alteromonadales*, *Oceanospirillales*, *Flavobacteriales*, and *Rhodospirillales*, are more abundant, which have higher growth rates and *rrn* copy numbers than free-living microbes⁹ and are also commonly found in coastal sediments as shown in this study. This is similar to the decrease of oligotrophs (e.g., SAR11) while enrichment of copiotrophs in deep ocean water (Figure S2), where the organic matter availability is higher due to particle deposition³².”

New figure: The impact of oligotrophs on the “*rrn* copy number - abundance” patterns in the Tara Oceans dataset. **a**, the *rrn* copy number distribution of the OTUs belonging to the abundant biosphere. **b**, OTUs’ *rrn* copy number of the abundant, intermediate, and rare biospheres

A3: Thanks a lot for clarifying.

A4: Thanks a lot, I’m satisfied with the current modification.

Response: we deeply appreciate your help in improving the manuscript!

清華大學
TSINGHUA UNIVERSITY

环境学院
SCHOOL OF ENVIRONMENT

B. Response to Reviewer #2 (Remarks to the Author)

In this revision, Dai and co-authors have done a good job at addressing the questions and comments previously raised by myself and the other reviewer. The authors have clarified sections of the abstract and text that were previously vague, and have provided additional info regarding particular methods, which is appreciated. The authors have also convincingly shown that the data and patterns observed are robust, with and without Sar11 in the Tara Ocean dataset analysis, as they indicated in Figure. S1. As mentioned before, this is an impressive and comprehensive dataset and I look forward to seeing this study in its final published version.

Response: Thanks for the encouragement! We deeply appreciate your help in improving the manuscript.

Reviewer comments, third round –

Reviewer #1 (Remarks to the Author):

Reply to A1 Response: Since you agreed with the comment, I would expect at least some improvement should be done, such as comparing the rrn copy numbers conditioning on the underlying phylogenetic structure.

Reply to A2 Response: Thanks a lot for all your efforts. However, the current re-analysis is focusing on comparing the rrn copy numbers of abundant, intermediate and rare biospheres in only oceanic waters, instead of across nutrient gradients of different environments. It has to mention that the abundant, intermediate and rare biospheres are defined or can be derived from the same water samples, therefore within samples, the rrn copy numbers don't reflect the measured nutrient availability. Actually, the higher within-sample variance raises another concern that how reliable the between-sample comparison is. To test the "hunger game" hypothesis, the authors are supposed to redo the comparison between different environments with biased taxa removed (those single rrn copy oligotrophs in oceanic samples), then test if the rrn copy numbers are still higher in sediment samples than in oceanic waters, and the variance has to be significantly higher than within-sample variance.

Reviewer #1 (Remarks to the Author):

Reply to A1 Response: *Since you agreed with the comment, I would expect at least some improvement should be done, such as comparing the *rrn* copy numbers conditioning on the underlying phylogenetic structure.*

Response: If we have understood correctly, the reviewer's concern is whether changes in community-level *rrn* copy numbers are still induced by nutrient availability after controlling variations of phylogenetic structure. Here, we have performed the partial Mantel tests, which show that both phylogenetic structure and nutrients have important, statistically significant influence on community-level *rrn* copy numbers. Because the metadata are available for only 4 datasets (see Table S1) in which variables of nutrient availability were measured, the partial Mantel tests were performed for each dataset separately. All of the results show that the effect of nutrients on community-level *rrn* copy number is significant when the effect of the phylogenetic structure is controlled (see Table below). Therefore, the results should be robust and convincing since the datasets vary substantially by sample types, environmental conditions, and spatial scales. For example, the Tara Oceans dataset includes global samples collected from different oceanic regions or water columns, and the Hangzhou Bay dataset includes regional samples collected from a land-sea environmental gradient covering ~ 100 km.

We have added the results as Table S4 in the Supporting Information. Accordingly, we have added sentences in lines 148-150 "Partial Mantel tests confirmed significant linkages between community-level *rrn* copy number and nutrients ($r = 0.130 \sim 0.602$, $P < 0.029$) when differences in the underlying phylogenetic structure were controlled (Table S4)." We have described the methodology in lines 428-432 "The linkages between community-level *rrn* copy number and nutrients or community phylogenetic structure were tested by partial Mantel tests using the function *mantel.partial* in the "vegan" package. Euclidean distance was calculated to

reveal differences in nutrient availability, and weighted Unifrac distance was calculated to reveal phylogenetic dissimilarity.”

Table Influence of nutrient availability and phylogenetic structure on community-level *rrn* copy number by partial Mantel tests^a

Datasets		Nutrients		Phylogenetic structure	
		r	P	r	P
Coastal sediment	Hangzhou Bay, China	0.220	0.026	0.751	0.001
	Coastal Sydney	0.406	0.001	0.266	0.001
	Coastal Mediterranean	0.602	0.001	-0.229	0.947
Ocean water	the Tara Oceans	0.130	0.029	0.483	0.001

^aChanges in the phylogenetic structure are measured by weighted Unifrac distance, and changes in nutrients availability are measured by Euclidean distance.

If Reviewer #1 thinks that we can use some help in how to compare the *rrn* copy numbers conditioning on the underlying phylogenetic structure, we will be grateful if the reviewer can explicitly recommend more data analysis methods.

Reply to A2 Response: *Thanks a lot for all your efforts. However, the current re-analysis is focusing on comparing the rrn copy numbers of abundant, intermediate and rare biospheres in only oceanic waters, instead of across nutrient gradients of different environments. It has to mention that the abundant, intermediate and rare biospheres are defined or can be derived from the same water samples, therefore within samples, the rrn copy numbers don't reflect the measured nutrient availability. Actually, the higher within-sample variance raises another concern that how reliable the between-sample comparison is. To test the "hunger game" hypothesis, the authors are supposed to redo the comparison between different environments with biased taxa removed (those single rrn copy oligotrophs in oceanic samples), then test if the rrn copy numbers are still higher in sediment samples than in oceanic waters, and the variance has to be significantly higher than within-sample variance.*

Response: We have compared the *rrn* copy numbers across different samples as suggested. When all OTUs are included, the *rrn* copy number of the abundant biosphere in ocean water is the lowest (panel **a**), and that of the rare biosphere is the highest (panel **b**), as compared to the sediment samples. When removing the single *rrn* copy OTUs in oceanic samples, the *rrn* copy number of the abundant biosphere in ocean water is only significantly lower than the sediments in Mission Bay and the Gulf of Mexico (panel **c**), while that of the rare biosphere is still significantly higher than the sediment samples (panel **d**). As the reviewer expected, the between-group variance of ocean water and coastal sediment is much larger than the within-group variance (see Table below) when the difference is significant ($P < 0.05$). These results indicate that single *rrn* copy OTUs in ocean water indeed contribute the most to the low values of the abundant biosphere.

In fact, 12.6% (3389/26961) of all the OTUs detected in ocean water have a single *rrn* copy with a sequence abundance of 11.04% ~ 63.73% in each sample, showing a higher prevalence of them as compared to OTUs with more *rrn* copies (2 ~ 15). Of the abundant biosphere OTUs in ocean water, 80.95% (85/105) have a single *rrn* copy with a sequence abundance of 1.23% ~ 26.75%, further demonstrating that they are favored in the oligotrophic ocean water. These data suggest the enrichment of single *rrn* copy OTUs in global ocean water, which per se supports the “hunger games” hypothesis that high *rrn* copy number is favored in copiotrophic environment but low *rrn* copy number is favored in the oligotrophic environment.

Nevertheless, as we have replied to Reviewer #2 in the earlier response letter, the *rrn* copy number of the abundant biosphere in ocean water is still significantly lower than the rare biosphere no matter whether OTUs with a single *rrn* copy are removed or not, also supporting the “hunger games” hypothesis.

We have added panels **a** and **b** below as Figure S1 in the Supporting Information, and added a sentence in lines 112-114 “Compared to the coastal sediments, the OTUs’ *rrn* copy numbers for the abundant biosphere in ocean water was significantly lower (Figure S1a), but that for the rare biosphere was significantly higher (Figure S1b).” If the reviewer thinks it necessary, we

will include the results and discussion of panels **c** and **d** in the revised manuscript, so that the readers would be more clear about the data structure. However, we do believe that panels **a** and **b** are more appropriate because the prevalence of OTUs with a single *rrn* copy in oceanic water is a result of natural selection^{1,2}. Therefore, they should be included in most data analyses.

Figure The *rrn* copy numbers of abundant and rare biosphere OTUs. All OTUs are included in panels **a** and **b**, while OTUs with a single *rrn* copy in the ocean water dataset are removed in panels **c** and **d**. Error bars represent the standard errors and lowercase letters above the bars indicate significant differences based on the LSD test after ANOVA (adjusted $P < 0.05$). In panels **a** and **c**, the bar representing Dois Rios estuary is missed because no OTU in the dataset is classified as abundant biosphere based on the criteria we defined.

Table Statistics of the comparisons between coastal sediments and ocean water samples

In comparison to ocean water samples	Within-group variance	Between-group variance	F	Critical value of F	P
Abundant biosphere (all OTUs)					
Mission Bay	0.7455	69.41	93.1	3.9243	<0.001
Hangzhou Bay	1.3144	84.71	64.45	3.9038	<0.001
the Gulf of Mexico	1.1403	191.1	167.6	3.8911	<0.001
Plymouth Harbor	0.8743	121.8	139.3	3.8984	<0.001
Coastal Mediterranean	0.7916	86.58	109.4	3.8829	<0.001
Coastal Sydney	0.7402	39.66	53.58	3.9064	<0.001
Rare biosphere (all OTUs)					
Mission Bay	3.8497	148	38.44	3.8418	<0.001
Hangzhou Bay	2.8315	5920	2091	3.8417	<0.001
the Gulf of Mexico	3.4675	1013	292.2	3.8418	<0.001
Plymouth Harbor	2.6190	4188	1599	3.8417	<0.001
Coastal Mediterranean	3.3470	1981	591.7	3.8418	<0.001
Coastal Sydney	1.9333	6221	3218	3.8415	<0.001
Abundant biosphere (remove single rrn copy OTUs)					
Mission Bay	1.2747	13.13	10.3	3.9934	0.002
Hangzhou Bay	1.9380	0.67	0.344	3.9361	0.559
the Gulf of Mexico	1.5278	10.74	7.026	3.9097	0.009
Plymouth Harbor	1.2244	3.06	2.502	3.9243	0.116
Coastal Mediterranean	0.9956	1.58	1.589	3.8951	0.209
Coastal Sydney	1.0920	2.01	1.843	3.9423	0.178
Rare biosphere (remove single rrn copy OTUs)					
Mission Bay	3.6670	397	108.1	3.8419	<0.001
Hangzhou Bay	2.5938	9255	3568	3.8417	<0.001
the Gulf of Mexico	3.2549	2202	676.7	3.8418	<0.001
Plymouth Harbor	2.4007	7486	3118	3.8417	<0.001
Coastal Mediterranean	3.1289	3708	1185	3.8418	<0.001
Coastal Sydney	1.8272	10999	6019	3.8415	<0.001

References:

1. Giovannoni SJ. SAR11 Bacteria: The Most Abundant Plankton in the Oceans. *Ann Rev Mar Sci* **9**, 231-255 (2017).
2. Giovannoni SJ, *et al.* Genome streamlining in a cosmopolitan oceanic bacterium. *Science* **309**, 1242-1245 (2005).

Reviewer comments, final round –

Reviewer #1 (Remarks to the Author):

A1:

Thanks a lot for your efforts, I'm satisfied with the current modification.

A2:

Thank you for performing this analysis. As we can see from this supplemental figure, the single copy OTUs in oceanic water have a great impact on the interpretation of the abundant biosphere but not the rare one. These are mostly oligotrophs (as the other reviewer argued earlier) that are dominant in the free-living size fraction of the Tara Oceans datasets. While those microbes with higher rrn copy numbers in the rare biosphere are copiotrophs that are abundant in the particle-attached size fractions. When the comparison is done against mainly oligotrophs, the low rrn copy number is expected. It's arguable or even opposite to what you're proposing if the comparison was done against particle attached fraction where microbes with high rrn copy number are not rare. So I insist on showing subplots c and d if this work will be published and urge the authors to make the size fractionation issue clear in the abstract and conclusion sections. It should also be extensively discussed as this could be cell/particle size-based bias rather than evolutionary selection.

清华大学
TSINGHUA UNIVERSITY

环境学院
SCHOOL OF ENVIRONMENT

Reviewer #1 (Remarks to the Author):

Reply to A1 Response: *Thanks a lot for your efforts, I'm satisfied with the current modification.*

Response: Thank you very much!

Reply to A2 Response: *Thank you for performing this analysis. As we can see from this supplemental figure, the single copy OTUs in oceanic water have a great impact on the interpretation of the abundant biosphere but not the rare one. These are mostly oligotrophs (as the other reviewer argued earlier) that are dominant in the free-living size fraction of the Tara Oceans datasets. While those microbes with higher *rrn* copy numbers in the rare biosphere are copiotrophs that are abundant in the particle-attached size fractions. When the comparison is done against mainly oligotrophs, the low *rrn* copy number is expected. It's arguable or even opposite to what you're proposing if the comparison was done against particle attached fraction where microbes with high *rrn* copy number are not rare. So I insist on showing subplots c and d if this work will be published and urge the authors to make the size fractionation issue clear in the abstract and conclusion sections. It should also be extensively discussed as this could be cell/particle size-based bias rather than evolutionary selection.*

Response: We have added the subplots c and d in Supplementary Figure 1 and removed the previous Supplementary Figure 2, which contained duplicated information. Accordingly, we have revised the sentences in lines 118-124 as “To assess how SAR11 and other OTUs with a single *rrn* copy affect *rrn* copy numbers, we removed those OTUs from the datasets. We showed that the *rrn* copy number of the abundant biosphere in ocean water was only significantly lower than the sediments in Mission Bay and the Gulf of Mexico (Supplementary Figure 1c). In contrast, that of the rare biosphere was still significantly higher than the sediment samples (Supplementary Figure 1d), indicating that single *rrn* copy OTUs in ocean water contributed the most to low *rrn* copy numbers of the abundant biosphere.” To make the size

清华大学
TSINGHUA UNIVERSITY

环境学院
SCHOOL OF ENVIRONMENT

fractionation issue clear, we have revised the sentence about ocean water community to be “pelagic zone of the global ocean” in the Abstract, and used “free-living” throughout the manuscript in lines 75, 236, 254, 313, and 365.

To acknowledge and discuss the limitations and alternative inferences of our results, we have revised the sentence in lines 236-244 as “Here, we show that it is also true in free-living marine bacterial communities (Figure 1 & 3, Supplementary Table 2). However, the particle-associated microbial communities have a quite different pattern where copiotrophs such as *Alteromonadales*, *Oceanospirillales*, *Flavobacteriales*, and *Rhodospirillales*, are more abundant, which have higher growth rates and *rrn* copy numbers than free-living microbes⁹ and are also commonly found in coastal sediments as shown in this study. This is similar to the decrease of oligotrophs (e.g., SAR11) and enrichment of copiotrophs in deep ocean water (Supplementary Figure 2), wherein the organic matter availability is higher due to particle deposition³².”